# Opinion Maximization in Social Networks by Modifying Internal Opinions

**Gengyu Wang, Runze Zhang, Zhongzhi Zhang**[*]
College of Computer Science and Artificial Intelligence
Fudan University
{gywang24, rzzhang24}@m.fudan.edu.cn, zhangzz@funda.edu.cn

## Abstract

Public opinion governance in social networks is critical for public health campaigns, political elections, and commercial marketing. In this paper, we addresse the problem of maximizing overall opinion in social networks by strategically modifying the internal opinions of a fixed number of nodes. Traditional matrix inversion methods suffer from prohibitively high computational costs, prompting us to propose two efficient sampling-based algorithms. Furthermore, we develop a deterministic asynchronous algorithm that exactly identifies the optimal set of nodes through asynchronous update operations and progressive refinement, ensuring both efficiency and precision. Extensive experiments on real-world datasets demonstrate that our methods outperform baseline approaches. Notably, our asynchronous algorithm delivers exceptional efficiency and accuracy across all scenarios, even in networks with tens of millions of nodes.

## 1 Introduction

Online social networks have fundamentally transformed the dissemination, evolution, and formation of opinions, serving as a powerful catalyst for accelerating and amplifying modern perspectives [1]. Compared to traditional communication methods, they facilitate faster, broader, and more decentralized information exchange, thereby enhancing the universality, criticality, and complexity of information propagation [2]. Within this intricate interplay between network structure and human behavior, the concept of overall opinion emerges as a key quantitative metric, representing the focal point of public sentiment on contentious issues [3]. This quantified equilibrium of public opinion has been applied in fields such as commercial marketing, political elections, and public health campaigns.

The optimization of overall opinions has garnered significant attention in recent times. Various methods have been explored to optimize collective opinions, including modifying resistance coefficients [4–6], adjusting expressed opinions [7], and altering network structures [8, 9]. Meanwhile, research [10] has highlighted the significant correlation between node topological positions and the evolution of global opinions, revealing that changes in the internal opinions of nodes can have a nonlinear amplification effect on opinion propagation. This provides the possibility of optimizing public opinion at low cost by modifying the internal opinions of key nodes.

In this paper, we address the following optimization problem: given a social network with $n$ nodes and $m$ edges (whether directed or undirected), along with an integer $k$, how can we strategically identify the $k$ nodes and modify their internal opinions to maximize the overall opinion? Existing exact solution methods require a time complexity of $O(n^3)$, rendering them impractical for large-scale networks. We propose two sampling approaches to approximate the solution, but these approaches face the challenge of balancing between extensive sampling requirements and accuracy. Inspired

---

[*]Corresponding author.

by the random walk interpretation, we further introduce a asynchronous update algorithm that exactly identifies the optimal set of nodes through asynchronous update operations and progressive refinement. We conducted extensive experiments on various real-world networks to evaluate algorithm performance. The experimental results demonstrate that all three proposed algorithms significantly outperform baseline methods in terms of effectiveness. Moreover, our asynchronous algorithm exhibits both high efficiency and exact precision, while maintaining excellent scalability for networks with tens of millions of nodes.

## 2 Related Work

We review the related literature from the following two perspectives, including modeling opinion dynamics and optimization problems in opinion dynamics.

**Opinion Dynamics Models.** Opinion dynamics has been the subject of intense recent research to model social learning processes in various disciplines [11–13]. These models capture the mechanisms and factors influencing opinion formulation, shedding light on understanding the whole process of opinion shaping and diverse phenomena taking place in social media. In the past decades, numerous relevant models have been proposed [14–18]. Among various existing models, the DeGroot model [19] and the Friedkin-Johnson (FJ) model [20] are two popular ones. After their establishment, the DeGroot model and the FJ model have been extended in a variety of ways [11, 21, 8, 22, 4], by incorporating different factors affecting opinion dynamics, such as peer pressure [23], susceptibility to persuasion [4, 8], and opinion leader [7]. Under the formalism of these models, some relevant quantities, properties and explanations have been broadly studied, including the equilibrium expressed opinions [24–26], sufficient condition for the stability [27], the average internal opinion [24], interpretations [28, 25], and so on.

**Optimization Problems in Opinion Dynamics.** Recently, several optimization problems related to opinion dynamics have been formulated and studied for different objectives. For example, a long line of work has been devoted to maximizing the overall opinion by using different strategies, such as identifying a fixed number of individuals and setting their expressed opinions to 1 [7], changing agent's initial opinions [29, 8, 30], as well as modifying individuals' susceptibility to persuasion [4–6]. [31] studies the problem of allocating seed users to opposing campaigns with a goal to maximize the expected number of users who are co-exposed to both campaigns. In additon, [32] studies the problem of balancing the information exposure. These studies have far-reaching implications in product marketing, public health campaigns, and political candidates. Another major and increasingly important focus of research is optimizing some social phenomena, such as maximizing the diversity [33, 34], minimizing conflict [35, 36], disagreement [37, 38, 9], and polarization [39, 38, 9].

## 3 Preliminaries

This section is devoted to a brief introduction to some useful notations and tools, in order to facilitate the description of problem formulation and algorithms.

**Notations.** We denote scalars in $\mathbb{R}$ by normal lowercase letters like $a, b, c$, sets by normal uppercase letters like $A, B, C$, vectors by bold lowercase letters like $\boldsymbol{a}, \boldsymbol{b}, \boldsymbol{c}$, and matrices by bold uppercase letters like $\boldsymbol{A}, \boldsymbol{B}, \boldsymbol{C}$. We use $\mathbf{1}$ to denote the vector of appropriate dimensions with all entries being ones, and use $\boldsymbol{e}_i$ to denote the $i^{\text{th}}$ standard basis vector of appropriate dimension. Let $\boldsymbol{a}^\top$ and $\boldsymbol{A}^\top$ denote, respectively, transpose of vector $\boldsymbol{a}$ and matrix $\boldsymbol{A}$. We write $A(i, j)$ to denote the entry at row $i$ and column $j$ of $\boldsymbol{A}$ and we use $\boldsymbol{a}(i)$ to denote the $i^{\text{th}}$ element of vector $\boldsymbol{a}$. Let $\boldsymbol{a}_{\max}$, $\boldsymbol{a}_{\min}$, $\bar{\boldsymbol{a}}$ and $\boldsymbol{a}_{\text{sum}}$ denote the maximum element, the minimum element, the mean of the elements in vector $\boldsymbol{a}$ and the sum of all elements in vector $\boldsymbol{a}$, respectively.

**Graph and Related Matrices.** Let $\mathcal{G} = (V, E)$ denote an directed graph with $n = |V|$ nodes and $m = |E|$ edges. The existence of $(v_i, v_j) \in E$ means that there is an edge from $v_i$ to $v_j$. In what follows, $v_i$ and $i$ are used interchangeably to represent node $v_i$, when it is clear from the context. For a node $v \in V$, the in-neighbors of $v$ are given by $N_{\text{in}}(v) = \{u | (u, v) \in E\}$, and the out-neighbors of $v$ are given by $N_{\text{out}}(v) = \{u | (v, u) \in E\}$. The connections of graph $\mathcal{G} = (V, E)$ are encoded in its adjacency matrix $\boldsymbol{A} = (a_{i,j})_{n \times n}$, with the element $a_{i,j}$ being 1 if $(v_i, v_j) \in E$ and 0 otherwise.

For a node $i \in V$, its out-degree $d_i^+$ is defined as $d_i^+ = \sum_{j=1}^{n} a_{i,j}$, and its in-degree $d_v^-$ is defined as $d_i^- = \sum_{j=1}^{n} a_{j,i}$. The diagonal degree matrix of $\mathcal{G}$ is defined as $\boldsymbol{D} = \mathrm{diag}(d_1^+, d_2^+, \ldots, d_n^+)$. We define $\boldsymbol{L} = \boldsymbol{D} - \boldsymbol{A}$ and $\boldsymbol{P} = \boldsymbol{D}^{-1}\boldsymbol{A}$ as the Laplacian matrix and the transition matrix of graph $\mathcal{G}$.

**Opinion Dynamic Model.** In this work, we adopt an opinion formation model introduced by the work of DeGroot [19] and Friedkin and Johnsen [40], which has been used in [4, 6, 24]. In this model, each agent $i$ is endowed with an internal opinion $s_i$ in $[0, 1]$, where 0 and 1 are polar opposites of opinions about a certain topic. Each agent also has a parameter that represents the susceptibility to persuasion, which we call the resistance coefficient $\alpha_i \in (0, 1]$. The internal opinion $s_i$ reflects the intrinsic position of the agent $i$ on a certain topic. A higher value on the resistance coefficient $\alpha_i$ means that the agent is less willing to conform to the opinions of the neighbors in the social network. According to the opinion dynamics model, the final opinion of each agent $i$ is a function of the social network, the set of internal opinions, and the resistance coefficients, determined by computing the equilibrium state of a dynamic opinion updating process. The social network is represented as a graph where edges capture influence relationships, with an edge from $i$ to $j$ indicating that agent $i$ is influenced by the expressed opinion of agent $j$. During the process of opinion evolution, the internal opinion $s_i$ remains constant, while the expressed opinion $z_i^t$ evolves at time $t + 1$ as follows:

$$z_i^{t+1} = \alpha_i s_i + (1 - \alpha_i) \cdot \frac{\sum_{j \in N_{\mathrm{out}}(i)} z_j^t}{d_i^+},$$

which can also be expressed in matrix form as $\boldsymbol{z}^{t+1} = \boldsymbol{R}\boldsymbol{s} + (\boldsymbol{I} - \boldsymbol{R})\boldsymbol{P}\boldsymbol{z}^t$. This dynamic converges to a unique equilibrium if $\alpha_i > 0$ for all $i \in V$ [14]. The equilibrium opinion vector $\boldsymbol{z}$ is the solution to a linear system of equations:

$$\boldsymbol{z} = (\boldsymbol{I} - (\boldsymbol{I} - \boldsymbol{R})\boldsymbol{P})^{-1}\boldsymbol{R}\boldsymbol{s}, \tag{1}$$

where $\boldsymbol{R} = \mathrm{Diag}(\alpha)$ is a diagonal matrix called resistance matrix and entry $\boldsymbol{R}(i, i)$ corresponds to $\alpha_i$. We call $\boldsymbol{z}(i)$ the expressed opinion of agent $i$. Let $\boldsymbol{M} = (\boldsymbol{I} - (\boldsymbol{I} - \boldsymbol{R})\boldsymbol{P})^{-1}\boldsymbol{R}$, we have $\boldsymbol{z} = \boldsymbol{M}\boldsymbol{s}$. Note that $\boldsymbol{M}$ is a row-stochastic matrix such that $\boldsymbol{M}\boldsymbol{1} = \boldsymbol{1}$.

## 4 Problem Formulation

An important quantity for opinion dynamics is the overall expressed opinion or the average expressed opinion at equilibrium, the optimization problem for which has been addressed under different constraints [7, 41, 4, 6, 8, 29, 30]. In this section, we propose a problem of maximizing overall expressed opinion in a graph, and introduce an exact algorithm optimally solving the problem.

**Overall Opinion and Structural Centrality.** For the opinion dynamic model in graph $\mathcal{G} = (V, E)$, the overall expressed opinion is defined as the sum $\boldsymbol{z}_{\mathrm{sum}}$ of expressed opinions $z_i$ of every node $i \in V$ at equilibrium. By Eq. (1), $\boldsymbol{z}_{\mathrm{sum}} = \boldsymbol{1}^\top \boldsymbol{M}\boldsymbol{s}$. Given the internal opinion vector $\boldsymbol{s}$ and the resistance matrix $\boldsymbol{R}$, we use $f(\boldsymbol{R}, \boldsymbol{s})$ to denote the overall expressed opinion. By definition,

$$f(\boldsymbol{R}, \boldsymbol{s}) = \boldsymbol{1}^\top \boldsymbol{z} = \boldsymbol{1}^\top \boldsymbol{M}\boldsymbol{s} = \sum_{u \in V} \sum_{v \in V} \boldsymbol{M}(u, v)\boldsymbol{s}(v). \tag{2}$$

Eq. (2) tells us that the overall expressed opinion $f(\boldsymbol{R}, \boldsymbol{s})$ is determined by three factors: the internal opinion and the resistance coefficient of every node, as well as the network structure characterizing interactions between nodes, all of which constitute the social structure of the opinion system. The first two are intrinsic property of each node, while the last one is a structure property of the network, both of which together determine the opinion dynamics system. Concretely, for the equilibrium expressed opinion $\boldsymbol{z}_u = \sum_{v \in V} \boldsymbol{M}(u, v)\boldsymbol{s}(v)$ of node $u$, $\boldsymbol{M}(u, v)$ indicates the convex combination coefficient or contribution of the internal opinion for node $v$. And the average value of the $v$-th column elements of $\boldsymbol{M}$, denoted by $\rho_v \triangleq \sum_{u \in V} \boldsymbol{M}(u, v)$, measures the contribution of the internal opinion of node $v$ to $f(\boldsymbol{R}, \boldsymbol{s})$. We call $\rho_v$ as the structural centrality [42] of node $v$ in the opinion dynamics model, since it catches the long-run structure influence of node $v$ on the overall expressed opinion. Note that matrix $\boldsymbol{M}$ is row stochastic and $0 \le \boldsymbol{M}(u, v) \le 1$ for any pair of nodes $u$ and $v$, $0 \le \rho_v \le n$ holds for every node $v \in V$, and $\sum_{v \in V} \rho_v = n$.

Using structural centrality, the overall expressed opinion $f(\boldsymbol{R}, \boldsymbol{s})$ is expressed as $f(\boldsymbol{R}, \boldsymbol{s}) = \sum_{v \in V} \rho_v \boldsymbol{s}(v)$, which shows that the overall expressed opinion $f(\boldsymbol{R}, \boldsymbol{s})$ is a convex combination of the internal opinions of all nodes, with the weight for $\boldsymbol{s}_v$ being the structural centrality $\rho_v$ of node $v$.

**Problem Statement.** As shown above, for a given graph $\mathcal{G} = (V, E)$, its node centrality remains fixed. For the FJ opinion dynamics model on $\mathcal{G} = (V, E)$ with internal opinion vector $s$ and resistance matrix $R$, if we choose a set $T \subseteq V$ of $k$ nodes and persuade them to change their internal opinions to 1, the overall equilibrium opinion, denoted by $f_T(R, s)$, will increase. It is clear that for $T = \emptyset$, $f_\emptyset(R, s) = f(R, s)$. Moreover, for two node sets $H$ and $T$, if $H \subseteq T \subseteq V$, then $f_T(R, s) \geq f_H(R, s)$. Then the problem OPINIONMAX of opinion maximization arises naturally: How to optimally select a set $T$ with a fixed number of $k$ nodes and change their internal opinions to 1, so that their influence on the overall equilibrium opinion is maximized. Let the vector $\mathbf{\Delta}$ be the potential influence vector, where $\mathbf{\Delta}(i) = \rho_i(1 - s(i))$ defines the potential influence of node $i$ on the growth of the overall equilibrium opinion. Mathematically, it is formally stated as follows.

---

**Problem 1.** *(OPINIONMAX) Given an unweighted graph $\mathcal{G} = (V, E)$, an internal opinion vector $s$, a resistance matrix $R$, and an integer parameter $k \ll n$, we aim to find the set $T \subset V$ with $|T| = k$ nodes, and change the internal opinions of these chosen $k$ nodes to 1, so that the overall equilibrium opinion is maximized. That is,*

$$T = \arg \max_{U \subset V, |U| = k} f_U(R, s) = \arg \max_{U \subset V, |U| = k} \sum_{i \in U} \mathbf{\Delta}(i). \tag{3}$$

---

Similarly, we can define the problem OPINIONMIN for minimizing the overall equilibrium opinion by optimally selecting a set $T$ of $k$ nodes and changing their internal opinions to 0. The goal of problem OPINIONMIN is to drive the overall equilibrium opinion $f_T(R, s)$ towards the polar value 0, while the goal of problem OPINIONMAX is to drive $f_T(R, s)$ towards polar value 1. Although the definitions and formulations of problems OPINIONMAX and OPINIONMIN are different, we can prove that they are equivalent to each other. In the sequel, we only consider the OPINIONMAX problem in this paper.

**Optimal Solution.** The most naive and straightforward method for solving Problem 1 involves directly computing $z$ by inverting the matrix $I - (I - R)P$, which has a complexity of $O(n^3)$. Identifying the top $k$ elements using a max-heap has a complexity of $O(n \log k)$. Therefore, the overall time complexity of the algorithm involving matrix inversion is $O(n^3)$. This impractical time complexity makes it infeasible for networks with only tens of thousands of nodes on a single machine. In the following sections, We propose a new interpretation and attempt to propose a new precise algorithm based on this explanation.

## 5 Sampling Methods

In this section, apart from the algebraic definition, we give two novel interpretations and propose corresponding sampling algorithms to approximately solve Problem 1.

**Random Walk-Based Algorithm.** Observing that the expression for the overall equilibrium opinion in Eq. (2) can be expanded as $\mathbf{1}^\top \sum_{i=0}^{\infty} ((I - R)P)^i R s$ via the Neumann series, we introduce the *absorbing random walk*. For a absorbing random walk starting from node $s$, at each step where the current node is $j$, the walk either (i) is absorbed by node $j$ with probability $\alpha_j$, or (ii) moves uniformly at random to a neighboring node with probability $1 - \alpha_j$, where the resistance coefficient $\alpha_j$ of node $j$ is represented as the absorption probability of the random walk at node $j$.

**Lemma 1.** *For an unweighted graph $\mathcal{G} = (V, E)$, let $p_i \in \mathbb{R}^{|V|}$ be the absorption probability vector of absorbing random walks starting at node $i \in V$. The structural centrality of node $v$ is $\rho_v = \sum_{i \in V} p_i(v)$.*

Leveraging this connection between structural centrality and termination probabilities, we propose a random walk-based algorithm RWB to efficiently estimate structural centrality for all nodes and compute an approximate solution to Problem 1. In algorithm RWB, we first simulate $N$ runs of the absorbing random walk $\{X_i\}_{i \geq 0}$, where each realization starts from a node uniformly chosen from $V$. We then estimate the structural centrality $\rho_j$ for each node $j \in V$ by scaling the empirical absorption frequency at $j$ by $n$.

**Lemma 2.** *Let $\mathcal{G} = (V, E)$ be an unweighted graph with internal opinion vector $s$, and resistance matrix $R$. For any error tolerance $\epsilon \in (0, 1)$, if algorithm RWB simulates $N = O\left(\frac{n}{\epsilon^2} \log n\right)$*

*independent random walks, then the estimated structural centrality $\hat{\rho}_i$ of any node $i \in V$ satisfies* $\Pr\left(|\hat{\rho}_i - \rho_i| \geq \epsilon\right) \leq \frac{1}{n}$.

**Theorem 1.** *Consider a graph $\mathcal{G} = (V, E)$ with internal opinion vector $\boldsymbol{s}$ and resistance matrix $\boldsymbol{R}$. Let $\alpha_{\min} = \min_{i \in V} R(i,i)$ and $\alpha_{\max} = \max_{i \in V} R(i,i)$. Under the error guarantee of Lemma 2, algorithm* RWB *achieves a time complexity of $O(\frac{\alpha_{\max}(1-\alpha_{\min})}{\epsilon^2 \alpha_{\min}^2} \cdot n \log n)$.*

We now establish that algorithm RWB provides provable approximation guarantees for OPINIONMAX. While Lemma 2 bounds the error of individual $\hat{\rho}_i$ estimates, the following result demonstrates that the *collective quality* of the selected set $\hat{T}$ is near-optimal:

**Corollary 1.** *Let $\hat{\rho}_i$ be the estimator of $\rho_i$ from algorithm* RWB *with absolute error parameter $\epsilon$, and $T^*$ be the optimal solution to Problem 1. For the set $\hat{T}$ consisting of the $k$ nodes achieving $\arg\max_{|T|=k} \sum_{i \in T} \hat{\rho}_i(1 - s_i)$, we have: $\sum_{i \in \hat{T}} \rho_i(1 - s_i) \geq \sum_{i \in T^*} \rho_i(1 - s_i) - 2k\epsilon$.*

**Forest Sampling Algorithm.** For matrix $\boldsymbol{Q} + \boldsymbol{L}$, where $\boldsymbol{Q}$ is a diagonal matrix, the elements of its inverse can be interpreted combinatorially in terms of spanning converging forests in a graph [43, 44]. In particular, by setting $\boldsymbol{Q} = \boldsymbol{D}(\boldsymbol{I} - \boldsymbol{R})^{-1}\boldsymbol{R}$, we can reformulate Eq. (2) in terms of the fundamental matrix, thereby establishing a direct correspondence between the structural centrality and spanning converging forests.

**Lemma 3.** *Let $\mathcal{F}$ be the set of all spanning converging forests of graph $\mathcal{G}$, and let $\mathcal{F}^{ij} \subseteq \mathcal{F}$ be the subset where nodes $i$ and $j$ are in the same converging tree rooted at node $i$. For a forest $F \in \mathcal{F}$, let $r(F)$ be the set of roots of $F$. Then the structural centrality of node $i$ is $\rho_i = \frac{\sum_{j \in V} \sum_{F \in \mathcal{F}^{ij}} \prod_{u \in r(F)} Q(u,u)}{\sum_{F \in \mathcal{F}} \prod_{u \in r(F)} Q(u,u)}$.*

This theoretical insight connects significantly with an extension of Wilson's algorithm. As shown in [30], Wilson's algorithm, based on loop-erased random walks, effectively simulates the probability distribution of rooted spanning trees when each node $i \in V$ is assigned a probability $\boldsymbol{Q}(i,i)$ of being the root. Building upon this theoretical framework, we propose an algorithm FOREST. The experiment in [30] demonstrates that the algorithm can maintain good effectiveness with a small number of samplings. More details are presented in Appendix B. The following theorem provide its time complexity:

**Theorem 2.** *When the number of samplings is $l$, the time complexity of algorithm* FOREST *is $O(\frac{1}{\alpha_{\min}} ln)$.*

## 6 Fast Exact-Selection Method via Asynchronous Updates

Sampling methods face challenges in identifying optimal size-$k$ node sets with maximal potential influence, as their sample complexity scales as $\epsilon^{-2}$. In this section, we present a deterministic asynchronous algorithm that provides rigorous error guarantees, enabling exact computation of the highest-influence node set without reliance on stochastic samplings. To maintain consistency, we present our derivation using unweighted graphs, noting that the corresponding algorithms can be readily extended to weighted graphs.

### 6.1 Asynchronous Update-Based Approximation

Let $\boldsymbol{r}^t$ denote the residual vector at time $t$, initialized as $\boldsymbol{r}^0 = \boldsymbol{1}$, and updated recursively via $\boldsymbol{r}^{t+1} = \boldsymbol{P}^\top(\boldsymbol{I} - \boldsymbol{R})\boldsymbol{r}^t$. By unrolling this recurrence, we obtain $\boldsymbol{r}^t = (\boldsymbol{P}^\top(\boldsymbol{I} - \boldsymbol{R}))^t \boldsymbol{1}$, which captures the distribution probability of a $t$-step absorbing random walk originating from a uniformly chosen node. This probabilistic interpretation motivates our deterministic algorithm employing asynchronous computation. The asynchronous paradigm provides two fundamental advantages: First, it enables local computation where each node's residual evolves independently based on its neighborhoods; second, it supports node-specific termination through local convergence monitoring of individual node states. These properties collectively overcome the synchronization constraints of global methods while eliminating the sampling overhead of probabilistic approaches.

**Global Influence Approximation.** We propose an efficient asynchronous algorithm to compute the potential influence vector $\boldsymbol{\Delta}$ for the OPINIONMAX problem. The algorithm maintains a residual vector $\boldsymbol{r}_a$ initialized to $\boldsymbol{1}$, an estimated potential influence vector $\hat{\boldsymbol{\Delta}}$ initialized to $\boldsymbol{0}$, and uses a

boundary vector $\boldsymbol{h} = \epsilon\mathbf{1}$ (which $\epsilon \in (0,1)$) as the termination criterion. At each iteration, for every node $i$ satisfying $\boldsymbol{r}_a(i) > \boldsymbol{h}(i)$, the algorithm performs three key operations: (i) distributing $(1-\alpha_i)/d_i^+ \cdot \boldsymbol{r}_a(i)$ to each out-neighbor's residual $\boldsymbol{r}_a(j)$ for $j \in N_{\text{out}}(i)$, (ii) accumulating $(1-s_i)\alpha_i \cdot \boldsymbol{r}_a(i)$ to the node's own potential influence estimate $\hat{\boldsymbol{\Delta}}(i)$, and (iii) resetting $\boldsymbol{r}_a(i)$ to 0. These updates are managed asynchronously through a first-in-first-out queue $Q$, processing nodes when they meet the residual threshold condition. The algorithm terminates when the residual condition $\boldsymbol{r}(v) \leq \boldsymbol{h}(v)$ holds for all nodes $v \in V$. Each push operation maintains strict locality by only accessing direct neighbors, while the asynchronous execution enables efficient computation of the potential influence scores needed for opinion maximization. The pseudo code is provided in Algorithm 1.

---

**Algorithm 1:** GLOBALINFAPPROX$(\mathcal{G}, \boldsymbol{R}, \boldsymbol{s}, \epsilon)$

**Input** : Graph $\mathcal{G} = (V, E)$, resistance matrix $\mathbf{R}$, internal opinion vector $\mathbf{s}$, error parameter $\epsilon$.
**Output** : Estimated potential influence vector $\hat{\boldsymbol{\Delta}}$ and residue vector $\boldsymbol{r}_a$.

1 **Initialize** : $\hat{\boldsymbol{\Delta}} = 0$; $\boldsymbol{r}_a = \mathbf{1}$
2 **while** $\exists v \in V$ *s.t.* $\boldsymbol{r}_a(v) > \epsilon$ **do**
3 $\quad$ $\hat{\boldsymbol{\Delta}}(v) = \hat{\boldsymbol{\Delta}}(v) + (1 - \boldsymbol{s}(v))\alpha_v \boldsymbol{r}_a(v)$
4 $\quad$ **for** *each* $u \in N_{out}(v)$ **do**
5 $\quad\quad$ $\boldsymbol{r}_a(u) = \boldsymbol{r}_a(u) + \frac{1-\alpha_v}{d_v^+}\boldsymbol{r}_a(v)$
6 $\quad$ $\boldsymbol{r}_a(v) = 0$
7 **return** $\hat{\boldsymbol{\Delta}}$, $\boldsymbol{r}_a$

---

The correctness of the algorithm relies on the relationship between the residual vector and the estimation error, which is guaranteed by the following lemma.

**Lemma 4.** *For any node $i \in V$ during the execution of Algorithm 1, the equality $\boldsymbol{\Delta}(i) - \hat{\boldsymbol{\Delta}}(i) = (1 - \boldsymbol{s}(i)) \cdot \boldsymbol{e}_i^\top \boldsymbol{M}^\top \boldsymbol{r}_a$ holds.*

Using Lemma 4, we can further establish the relative error guarantees for the results returned upon termination of the Algorithm 1, as shown in the following lemma.

**Lemma 5.** *For any parameter $\epsilon \in (0, 1)$, the estimator $\hat{\boldsymbol{\Delta}}$ returned by Algorithm 1 satisfies the following relation: $(1 - \epsilon)\boldsymbol{\Delta}(v) \leq \hat{\boldsymbol{\Delta}}(v) \leq \boldsymbol{\Delta}(v), \forall v \in V$.*

**Targeted Node Refinement.** Algorithm 1 provides solutions with rigorous relative error guarantees. As established in Lemma 8, these precise error bounds allow us to determine whether a given node must necessarily belong to the size-$k$ set with maximal potential influence. Furthermore, when the error tolerance $\epsilon$ becomes sufficiently small, we can completely identify the exact size-$k$ node set that maximizes opinion influence. However, since each error threshold setting generates a distinct candidate set of boundary nodes, repeatedly computing global error bounds through Algorithm 1 would incur unnecessary computational overhead. This motivates our key optimization: Can we focus computations exclusively on a reduced candidate set identified by Algorithm 1 to improve efficiency?

---

**Algorithm 2:** TARGETEDNODEREFINE$(\mathcal{G}, \boldsymbol{r}_a, \boldsymbol{r}^0, \epsilon)$

**Input** : Graph $\mathcal{G} = (V, E)$, forward residual vector $\boldsymbol{r}_a$, initial residual vector $\boldsymbol{r}^0$, error parameter $\epsilon$.
**Output** : Estimator $\tilde{\Delta}$ and residue vector $\boldsymbol{r}_s$.

1 **Initialize** : $\tilde{\Delta} = 0$; $\boldsymbol{r}_s = \boldsymbol{r}^0$
2 **while** $\exists v \in V$ *s.t.* $\boldsymbol{r}_s(v) > \epsilon\alpha_v$ **do**
3 $\quad$ $\tilde{\Delta} = \tilde{\Delta} + \boldsymbol{r}_a(v) \cdot \boldsymbol{r}_s(v)$
4 $\quad$ **for** *each* $u \in N_{in}(v)$ **do**
5 $\quad\quad$ $\boldsymbol{r}_s(u) = \boldsymbol{r}_s(u) + \frac{1-\alpha_u}{d_u^+}\boldsymbol{r}_s(v)$
6 $\quad$ $\boldsymbol{r}_s(v) = 0$
7 **return** $\tilde{\Delta}$, $\boldsymbol{r}_s$

---

The theoretical foundation comes from Lemma 4, which provides the exact decomposition for all nodes in the network. This decomposition directly motivates Algorithm 2, designed to approximate the influence propagation term $(1 - s(i)) \cdot e_i^\top M^\top r_a$ for any given node $i$. The algorithm initializes with estimate $\tilde{\Delta} = 0$ and residual vector $r_s = r^0$, using $h = \epsilon R 1$ as the boundary vector. For each node $v$ satisfying $r_s(v) > h(v)$, the algorithm updates $\tilde{\Delta}$ by adding $r_a(v) \cdot r_s(v)$ and propagating residuals to in-neighbors before resetting $r_s(v) = 0$. The process repeats until $r_s(v) \leq h(v)$ for all nodes.

While Algorithm 2's asynchronous operations do not maintain the random walk interpretation, its correctness is established through the following analysis.

**Lemma 6.** *When Algorithm 2 is initialized with $r^0 = \alpha_v(1 - s(v))e_v$, the equality $\Delta(v) - (\hat{\Delta}(v) + \tilde{\Delta}) = r_a^\top M R^{-1} r_s$ holds throughout execution.*

Building upon Lemma 6, we can derive the following absolute error bound when specific initialization conditions are met.

**Lemma 7.** *For Algorithm 2 with initial residual $r^0 = \alpha_v(1 - s(v))e_v$, given any parameter $\epsilon \in (0, 1)$, the estimator $\tilde{\Delta}$ satisfies the absolute error bound: $0 < \Delta(v) - (\hat{\Delta}(v) + \tilde{\Delta}) \leq \epsilon \cdot (r_a)_{\text{sum}}$.*

### 6.2 Fast Exact-Selection Algorithm

Let $T \subseteq V$ be the set of nodes guaranteed to be contained in the optimal size-$k$ node set with maximal value of sequence $a$, and let $C \subseteq V \setminus T$ denote the candidate nodes that may belong to this optimal set when considering estimation errors. Given an estimator $\hat{a}$ with uniform error bounds (either absolute or relative), we can formally characterize these sets through the following lemma.

**Lemma 8.** *Let $\hat{a}$ approximate $a$ with uniform error $0 < a(i) - \hat{a}(i) \leq \epsilon_a$ or $0 < a(i) - \hat{a}(i) \leq \epsilon_b a(i)$ for all $i \in V$, and let $\hat{a}_{}$ denote the $i$-th largest value in $\hat{a}$. Then,*

- *Element $i \in T$ if either $\hat{a}(i) \geq \hat{a}_{<k+1>} + \epsilon_a$ or $\hat{a}(i) \geq \hat{a}_{<k+1>}/(1 - \epsilon_b)$;*

- *Element $i \in C$ if $i \notin T$ and either $\hat{a}(i) \geq \hat{a}_{<k>} - \epsilon_a$ or $\hat{a}(i) \geq \hat{a}_{<k>}(1 - \epsilon_b)$.*

Let $k_{\text{gap}}$ denote the difference between the $k$-th and $(k + 1)$-th largest values in the true potential influence vector $\Delta$. When the absolute error $\epsilon < k_{\text{gap}}$, we can guarantee exact identification of the optimal size-$k$ node set. However, since $k_{\text{gap}}$ is typically unknown a priori, we employ an iterative refinement approach that progressively tightens the error bound $\epsilon$ across successive iterations. This process leverages residual vectors from previous iterations as initial conditions, thereby reducing recomputation overhead through warm-start optimization. The algorithm operates through two computational phases. First, global relative error bounds are applied to identify a small candidate set $C$. Then, Algorithm 2 computes absolute error guarantees specifically for nodes in $C$ while

---

**Algorithm 3:** MAXINFLUENCESELECTOR$(\mathcal{G}, R, s, \epsilon, k)$

**Input** : Graph $\mathcal{G} = (V, E)$, opinions $s$, parameter $k$.
**Output** : Optimal node set $T$.

1 **Initialize** : $T = \emptyset; C = V$
2 $\hat{\Delta}, r_a = $ GLOBALINFAPPROX$(\mathcal{G}, R, s, \epsilon)$
3 Update $T$ and $C$ via Lemma 8 with $\{\hat{\Delta}(v)\}_{v \in V}$ and error $\epsilon$
4 Initialize $r_v = \alpha_v(1 - s_v) \cdot e_v, \forall v \in C$
5 **for** $\epsilon' = 1, \frac{1}{2}, \ldots, \frac{1}{2n}, \ldots$ **do**
6    **for** $v \in C$ **do**
7      $\tilde{\Delta}, r_v = $ TARGETEDNODEREFINE$(\mathcal{G}, r_a, r_v, \epsilon'/(r_a)_{\text{sum}})$
8      $\hat{\Delta}(v) = \hat{\Delta}(v) + \tilde{\Delta}$
9    Update $T$ and $C$ via Lemma 8 with $\{\hat{\Delta}(v)\}_{v \in C}$ and error $\epsilon'$
10    **if** $|T| = k$ **then**
11      **return** $T$

---

progressively shrinking the candidate set size. The procedure terminates when the error bound satisfies $\epsilon < k_{\text{gap}}$ and the candidate set becomes empty, at which point we obtain the exact optimal size-$k$ set with maximal potential influence. The pseudo code is provided in Algorithm 3.

Theoretical guarantees of this approach are established in the following theorem, which provides an upper bound on the time complexity.

**Theorem 3.** *For sufficiently small $\epsilon$, the upper bound on the time complexity of Algorithm 3 is* $O(\frac{d_{\max}^+ n}{\alpha_{\min}} \log \frac{1}{\epsilon} + \frac{m}{k_{gap} \cdot \alpha_{\min}})$.

# 7 Experiments

In this section, we experimentally evaluate our proposed three algorithms. Additional experimental results and analyses are presented in Appendix C.

**Machine.** Our extensive experiments were conducted on a Linux server equipped with 28-core 2.0GHz Intel(R) Xeon(R) Gold 6330 CPU and 1TB of main memory. All the algorithms we proposed are implemented in *Julia v1.10.7* using single-threaded execution.

Table 1: Datasets

|  | DBLP | Google | YoutubeSnap | Pokec | Flixster | LiveJournal | Twitter | SinaWeibo |
|---|---|---|---|---|---|---|---|---|
| Nodes | 317,080 | 875,713 | 1,134,890 | 1,632,803 | 2,523,386 | 4,847,571 | 41,652,230 | 58,655,849 |
| Edges | 1,049,866 | 5,105,039 | 2,987,624 | 30,622,564 | 7,918,801 | 68,993,773 | 1,468,365,182 | 261,321,071 |
| $d_{\max}^+$ | 343 | 456 | 28,754 | 8,763 | 1,474 | 20,296 | 770,155 | 278,491 |
| Type | undirected | directed | undirected | directed | undirected | directed | directed | undirected |

**Datasets and Metrics.** We use 8 benchmark datasets that are obtained from the Koblenz Network Collection [45], SNAP [46] and Network Repository [47]. Table 1 summarizes the key characteristics of the networks used in our experiments, including network name, number of nodes, number of edges, maximum out-degree, and network type. The networks are listed in ascending order based on node count. A more detailed description of the datasets is provided in Appendix D. On each dataset, we employ the algorithm GLOBALINFAPPROX with a relative error parameter $10^{-12}$ to compute the ground-truth structural centrality scores. This ensures that each ground-truth value has at most $10^{-12}$ relative error. We evaluate the maximum influence node selection problem for varying set sizes $k \in \{1, 2, 4, \ldots, 1024\}$. We evaluate the accuracy of each method using three metrics: overall opinion, along with two standard ranking measures, *precision* and *Normalized Discounted Cumulative Gain (NDCG)* [48]. We initialize internal opinions with uniform distribution as their configuration does not significantly affect experimental results.

**Methods.** We present numerical results to evaluate the performance of our proposed algorithms, RWB, FOREST and MAXINFLUENCESET against existing baselines. For consistency, we abbreviate MAXINFLUENCESET as MIS in the following text. In all experiments, we set the absolute error parameters of RWB to $10^{-2}$. For algorithm FOREST, we set the number of samplings $l = 4000$. We set the initial error parameter of algorithm MIS to $10^{-3}$. To further demonstrate the effectiveness of our approach, we compare against five widely-used benchmark algorithms: TOPRANDOM, TOPDEGREE, TOPCLOSENESS, TOPBETWEENNESS, and TOPPAGERANK [49] across all networks.

**Resistance Coefficient Distributions.** In our experiments, resistance coefficient are generated using three distinct distributions: uniform, normal, and exponential. They are abbreviated as Unif., Norm., and Exp. in the figures. We set the minimum value of the resistance coefficient $\alpha_{\min} = 0.01$ because a zero resistance coefficient would lead to non-convergent results. For the uniform distribution, each node $i$ is assigned an opinion $s_i$ uniformly sampled from the interval $[\alpha_{\min}, 1]$. For the normal distribution, each node $i$ is assigned a sample $z_i$ drawn from the standard normal distribution $z_i \sim \mathcal{N}(0, 1)$. These values are then normalized to the interval $[\alpha_{\min}, 1]$. For the exponential distribution, we generate $n$ positive numbers $x$ using the probability density function $f(x) = e^{x_{\min}} e^{-x}$, where $x_{\min} > 0$. These values are similarly normalized to the range $[\alpha_{\min}, 1]$ to represent resistance coefficients.

**Efficiency.** Table 2 compares the execution time of three algorithms: RWB, FOREST, and our proposed MIS (with $k = 64$), under various resistance coefficient distributions. The direct matrix inversion based EXACT algorithm was excluded from the comparison as it failed to complete

Table 2: Running time with $k = 64$.

| Name | Time(s) | | | | | | | | |
|---|---|---|---|---|---|---|---|---|---|
| | RWB | | | FOREST | | | MIS | | |
| | Unif. | Norm. | Exp. | Unif. | Norm. | Exp. | Unif. | Norm. | Exp. |
| DBLP | 120.61 | 101.72 | 703.31 | 66.02 | 58.65 | 104.41 | 0.28 | 0.51 | 1.81 |
| Google | 344.02 | 326.83 | 1368.66 | 162.87 | 154.62 | 223.67 | 0.39 | 0.40 | 2.91 |
| YoutubeSnap | 896.66 | 900.97 | 10495.42 | 196.53 | 208.92 | 312.03 | 1.03 | 0.87 | 6.85 |
| Pokec | 1037.87 | 1028.07 | 6562.19 | 467.81 | 493.82 | 784.69 | 3.72 | 3.21 | 24.68 |
| Flixster | 2342.12 | 2142.68 | - | 326.52 | 321.75 | 376.91 | 2.59 | 2.14 | 13.56 |
| LiveJournal | 3275.20 | 2580.90 | - | 1454.37 | 1472.17 | 2147.34 | 8.61 | 13.17 | 56.49 |
| Twitter | - | - | - | 12115.80 | 13230.32 | 13852.17 | 279.73 | 270.61 | 1841.55 |
| SinaWeibo | - | - | - | 10538.27 | 10749.41 | 11743.13 | 156.83 | 171.75 | 1712.25 |

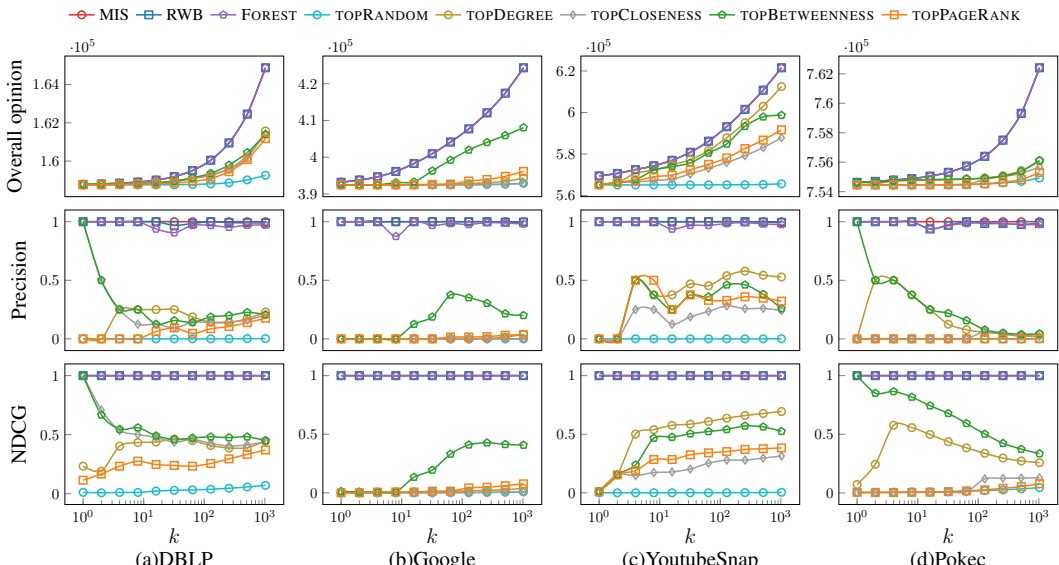

Figure 1: Performance of algorithms in accuracy across four smaller graphs.

even on the smallest DBLP network within the 6-hour time limit. Similarly, RWB executions exceeding 6 hours on large-scale networks with strict resistance distributions were terminated due to impractical runtime requirements. In contrast, both FOREST and MIS demonstrated robust scalability, successfully processing networks with tens of millions of nodes across all distribution scenarios. The data reveals that FOREST consistently achieves faster execution than RWB, while MIS outperforms both algorithms by a substantial margin in all test conditions. Notably, the execution time of RWB and MIS shows considerable sensitivity to resistance coefficient distributions, whereas FOREST maintains relatively stable performance in most cases despite distribution variations. As shown in Table 2, the excellent efficiency of our MIS algorithm can be easily applied in large-scale networks containing tens of millions of nodes, whether directed or undirected. This performance advantage makes MIS a feasible solution for large-scale network analysis tasks in the real world.

**Accuracy.** Figures 1 and 2 presents a comprehensive evaluation of opinion optimization performance, precision, and NDCG scores for all methods under uniformly distributed resistance coefficients. Due to excessively long running times, we omitted the performance of the RWB algorithm on the Twitter and SinaWeibo networks in Figure 2. Our experimental evaluation reveals that the proposed algorithms (RWB, FOREST, and MIS) consistently outperform all baseline methods across all evaluation metrics. This superior performance demonstrates the effectiveness of our approach in opinion optimization tasks. The result shows that while RWB and FOREST exhibit show measurable variations in precision under certain conditions, these fluctuations do not substantially affect their overall opinion optimization performance. The robustness of these algorithms across

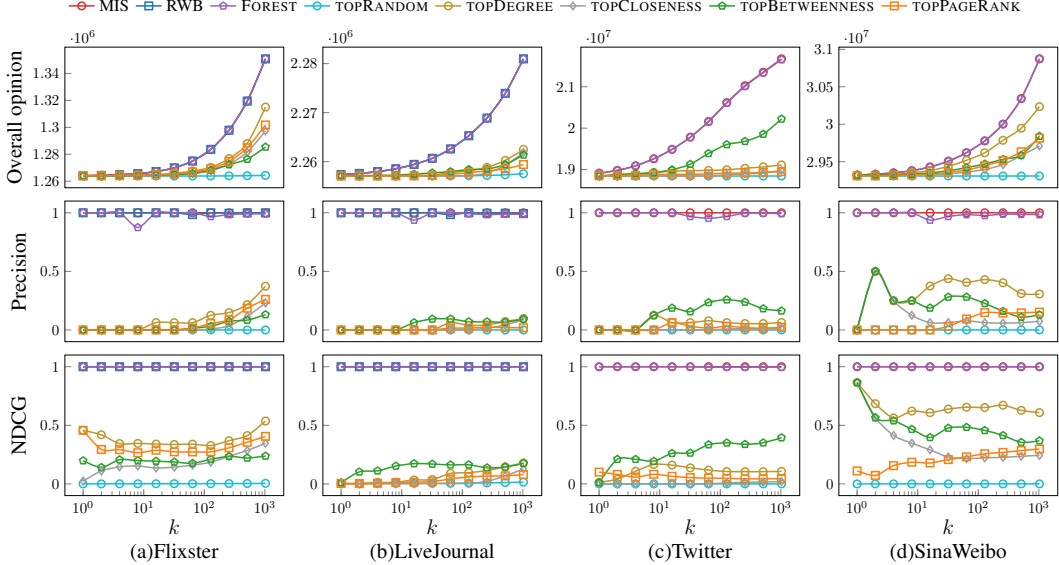

Figure 2: Performance of algorithms in accuracy across four larger graphs.

different network structures further confirms their practical utility. Notably, the MIS algorithm achieves perfect computational precision (exactly 1) while maintaining NDCG scores approaching 1, indicating both optimal opinion optimization results and accurate sequence ordering. When combined with its exceptional efficiency demonstrated in Table 2, these results further underscore the superiority of the MIS algorithm in both accuracy and computational performance.

## 8 Limitations

Despite its advantages, our MIS algorithm has certain limitations. As formally established in Theorem 3, the algorithm remains sensitive to resistance coefficients—particularly exhibiting prolonged runtime when these values are small. Furthermore, in exact optimization scenarios where boundary nodes contribute equally, the algorithm may encounter termination issues, though such cases are practically negligible. We note that this edge case can be effectively addressed by relaxing constraint conditions.

## 9 Conclusion

This paper proposes novel solutions for optimizing overall opinions in social networks by modifying the internal opinions of key nodes. As traditional matrix inversion methods face computational limitations in large-scale networks, we introduce two sampling-based algorithms. Building upon a random walk interpretation, we further develop a exact asynchronous update algorithm. This deterministic asynchronous approach provides guaranteed error bounds, leveraging asynchronous update operations and progressive refinement to efficiently and exactly identify nodes with the greatest potential influence. Extensive experiments demonstrate that compared to baseline methods and our sampling approaches, this method achieves superior efficiency and accuracy while effectively scaling to networks with tens of millions of nodes. Future research directions include extensions to dynamic network configurations and multi-opinion optimization scenarios.

## Acknowledgments

This work was supported by the National Natural Science Foundation of China (No. 62372112).

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

# Appendix

## A Omitted Proofs

### A.1 Proof of Lemma 1

According to Neumann series, we have $\boldsymbol{M} = (\boldsymbol{I} - (\boldsymbol{I} - \boldsymbol{R})\boldsymbol{P})^{-1}\boldsymbol{R} = \sum_{t=0}^{\infty} ((\boldsymbol{I} - \boldsymbol{R})\boldsymbol{P})^t \boldsymbol{R}$. Let $\boldsymbol{p}_i = \boldsymbol{e}_i^\top \sum_{t=0}^{\infty} ((\boldsymbol{I} - \boldsymbol{R})\boldsymbol{P})^t \boldsymbol{R}$ represent the absorption probability vector of absorbing random walks starting from node $i$. Hence, we have:

$$\rho_v = \sum_{i \in V} \boldsymbol{M}(i, v) = \sum_{i \in V} \boldsymbol{e}_i^\top \sum_{t=0}^{\infty} ((\boldsymbol{I} - \boldsymbol{R})\boldsymbol{P})^t \boldsymbol{R} \boldsymbol{e}_v = \sum_{i \in V} \boldsymbol{p}_i(v).$$

### A.2 Proof of Lemma 2

Algorithm RWB simulates $N$ independent runs of the absorbing random walk $\{X_i\}_{i=1}^N$, where:

$$X_i = \begin{cases} n, & \text{if the } i\text{-th walk is absorbed by } v, \\ 0, & \text{otherwise.} \end{cases}$$

The estimator $\hat{\rho}_v = \frac{1}{N} \sum_{i=1}^N X_i$ has the following properties:

$$\mathbb{E}[\hat{\rho}_v] = \frac{1}{N} \sum_{i=1}^N \mathbb{E}[X_i] = n \cdot \frac{1}{n} \sum_{u \in V} \boldsymbol{p}_u(v) = \rho_v.$$

$$\text{Var}[\hat{\rho}_v] = \frac{1}{N^2} \sum_{i=1}^N \text{Var}[X_i] = \frac{1}{N^2} \sum_{i=1}^N \left( \mathbb{E}[X_i^2] - \mathbb{E}[X_i]^2 \right)$$

$$= \frac{1}{N^2} \sum_{i=1}^N \left( n^2 \cdot \frac{\rho_v}{n} - \rho_v^2 \right) = \frac{\rho_v(n - \rho_v)}{N} \leq \frac{n^2}{4N}.$$

We now apply the following Chernoff bound for bounded variables.

**Lemma 9** (Chernoff Bound). *Let $X_i (1 \leq i \leq N)$ be independent random variables satisfying $X_i \leq E[X_i] + M$ for $1 \leq i \leq N$. Let $X = \frac{1}{N} \sum_{i=1}^N X_i$. Assume that $E[X]$ and $Var[X]$ are respectively the expectation and variance of $X$. Then we have*

$$\Pr(|X - E[X]| \geq \lambda) \leq 2 \exp(-\frac{\lambda^2 N}{2 Var[X] + 2M\lambda/3}).$$

Applying this lemma with $M = n$ and $\lambda = \epsilon$, we obtain

$$\Pr\left(|\hat{\rho}_v - \rho_v| \geq \epsilon\right) \leq 2 \exp\left(-\frac{\epsilon^2 N}{2 \cdot \frac{n^2}{4N} + 2n\epsilon/3}\right) = 2 \exp\left(-\frac{\epsilon^2 N}{\frac{n^2}{2N} + 2n\epsilon/3}\right).$$

When $N = O\left(\frac{n}{\epsilon^2} \log n\right)$, we have $\Pr\left(|\hat{\rho}_v - \rho_v| \geq \epsilon\right) \leq \frac{1}{n}$.

## A.3 Proof of Theorem 1

We first prove the expected length of a random walk. Since $\sum_{l=1}^{\infty} l \cdot x^{l-1} = \frac{1}{(1-x)^2}$ for $|x| < 1$, we have

$$\mathbb{E}[\text{Walk length starting from node } u] = \sum_{l=0}^{\infty} l \cdot \Pr(\text{Walk length starting from node } u \text{ is } l)$$

$$= \sum_{l=0}^{\infty} l \cdot e_u^{\top} \left((I - R)P\right)^l R\mathbf{1} \le \sum_{l=0}^{\infty} l(1 - \alpha_{\min})^l \alpha_{\max}$$

$$= \alpha_{\max}(1 - \alpha_{\min}) \sum_{l=1}^{\infty} l(1 - \alpha_{\min})^{l-1}$$

$$= \frac{\alpha_{\max}(1 - \alpha_{\min})}{\alpha_{\min}^2}.$$

According to Lemma 1, we have $N = \frac{n}{\epsilon^2} \log n$, hence the upper bound of time complexity is $O(\frac{\alpha_{\max}(1-\alpha_{\min})}{\epsilon^2 \alpha_{\min}^2} \cdot n \log n)$.

## A.4 Proof of Lemma 3

We begin by establishing the representation $M = (Q + L)^{-1}Q$. Starting from the definition of $M$, we derive

$$M = (I - (I - R)P)^{-1}R = (I - (I - R)D^{-1}A)^{-1}R = (D(I - R)^{-1} - A)^{-1}D(I - R)^{-1}R$$

$$= (D\sum_{i=0}^{\infty} R^i - A)^{-1}Q = (D\sum_{i=0}^{\infty} R^i \cdot R + D - A)^{-1}Q$$

$$= (D(I - R)^{-1}R + L)^{-1}Q = (Q + L)^{-1}Q.$$

To prove Lemma 3, we invoke a result from [44] concerning spanning converging forests:

**Lemma 10** ([44])**.** *Let $L_{-S}$ be the matrix obtained from $L$ by deleting the rows and columns corresponding to the nodes in $S \subseteq V$, and let $\mathcal{F}_S$ be the set of spanning converging forests of graph $\mathcal{G}$ with $|S|$ components that diverge from the nodes of $S$. Then, the determinant $\det(L_{-S}) = \sum_{F \in \mathcal{F}_S} w(F)$, where $w(F)$ represents the product of edge weights in forest $F$. Furthermore, for any nodes $i, j \in V \setminus S$, the $(i,j)$-cofactor $L_{-S}^{ij} = \sum_{F \in \mathcal{F}_{S \cup \{i\}}^{ij}} w(F)$.*

Let $L_q = M^{-1} = Q^{-1}(L + Q)$, We analyze $\det(L_q)$ as follows:

$$\det(L_q) = \det(Q^{-1})\det(L + Q) = \frac{1}{\prod_{v \in V} Q(v,v)} \det(L + Q)$$

$$= \frac{1}{\prod_{v \in V} Q(v,v)} \sum_{t=0}^{n} \sum_{\substack{S \subseteq V \\ |S|=t}} \det(L_{-S}) \prod_{u \in S} Q(u,u)$$

$$= \frac{1}{\prod_{v \in V} Q(v,v)} \sum_{t=0}^{n} \sum_{\substack{S \subseteq V \\ |S|=t}} \sum_{F \in \mathcal{F}_S} w(F) \prod_{u \in S} Q(u,u)$$

$$= \frac{1}{\prod_{v \in V} Q(v,v)} \sum_{S \subseteq V} \sum_{F \in \mathcal{F}_S} w(F) \prod_{u \in S} Q(u,u)$$

$$= \frac{1}{\prod_{v \in V} Q(v,v)} \sum_{F \in \mathcal{F}} w(F) \prod_{u \in r(F)} Q(u,u).$$

For the (i,j)-cofactor $\boldsymbol{L}_q^{ij}$, we similarly obtain

$$\det(\boldsymbol{L}_q^{ij}) = \det((\boldsymbol{Q}^{-1})^{ii}) \det((\boldsymbol{L}+\boldsymbol{Q})^{ij}) = \frac{Q(i,i)}{\prod_{v\in V} Q(v,v)} \det((\boldsymbol{L}+\boldsymbol{Q})^{ij})$$

$$= \frac{Q(i,i)}{\prod_{v\in V} Q(v,v)} \sum_{t=0}^{n-1} \sum_{\substack{S\subseteq V\setminus\{i,j\} \\ |S|=t}} \det(\boldsymbol{L}_{-S}^{ij}) \prod_{u\in S} \boldsymbol{Q}(u,u)$$

$$= \frac{Q(i,i)}{\prod_{v\in V} Q(v,v)} \sum_{t=0}^{n-1} \sum_{\substack{S\subseteq V\setminus\{i,j\} \\ |S|=t}} \sum_{F\in\mathcal{F}_{S\cup\{i\}}^{ij}} w(F) \prod_{u\in S} \boldsymbol{Q}(u,u)$$

$$= \frac{Q(i,i)}{\prod_{v\in V} Q(v,v)} \sum_{S\subseteq V\setminus\{i,j\}} \sum_{F\in\mathcal{F}_{S\cup\{i\}}^{ij}} w(F) \prod_{u\in S} \boldsymbol{Q}(u,u)$$

$$= \frac{1}{\prod_{v\in V} Q(v,v)} \sum_{F\in\mathcal{F}^{ij}} w(F) \prod_{u\in r(F)} \boldsymbol{Q}(u,u).$$

Since all edge weights in $\mathcal{G}$ are 1, the inverse $\boldsymbol{L}_q^{-1}$ simplifies to

$$\boldsymbol{L}_q^{-1}(i,j) = \frac{\det(\boldsymbol{L}_q^{ji})}{\det(\boldsymbol{L}_q)} = \frac{\sum_{F\in\mathcal{F}^{ji}} \prod_{u\in r(F)} Q(u,u)}{\sum_{F\in\mathcal{F}} \prod_{u\in r(F)} Q(u,u)}.$$

Finally, as $\rho_i = \sum_{v\in V} \boldsymbol{M}(v,i) = \sum_{v\in V} \boldsymbol{L}_q^{-1}(v,i)$, the expression for node $i$'s structural centrality $\rho_i$ follows

$$\rho_i = \sum_{j\in V} \frac{\sum_{F\in\mathcal{F}^{ij}} \prod_{u\in r(F)} Q(u,u)}{\sum_{F\in\mathcal{F}} \prod_{u\in r(F)} Q(u,u)}.$$

This completes the proof of Lemma 3.

## A.5  Proof of Theorem 2

The time complexity of the algorithm depends on the times nodes are visited in loop-erased random walks. Thus, the time complexity of FOREST is

$$l \cdot \sum_{v\in V} \sum_{i=0}^{\infty} ((\boldsymbol{I}-\boldsymbol{R})\boldsymbol{P})^i(v,v) = l \cdot \mathrm{Tr}((\boldsymbol{I}-(\boldsymbol{I}-\boldsymbol{R})\boldsymbol{P})^{-1}) = l \cdot \mathrm{Tr}(\boldsymbol{M}\boldsymbol{R}^{-1}) \leq \frac{l}{\alpha_{\min}} \mathrm{Tr}(\boldsymbol{M}).$$

Since $\boldsymbol{M}(i,j) \leq 1$ for any $i,j \in V$, the final time complexity is obtained as $O(\frac{1}{\alpha_{min}} ln)$.

## A.6  Proof of Corollary 1

Define the *ground-truth utility* of a set $T$ as $U(T) = \sum_{i\in T} \rho_i(1-s_i)$ and the *estimated utility* as $\hat{U}(T) = \sum_{i\in T} \hat{\rho}_i(1-s_i)$.

By the error condition $|\rho_i - \hat{\rho}_i| \leq \epsilon$, for any set $T$ with $|T| = k$, we have:

$$\left| \hat{U}(T) - U(T) \right| = \left| \sum_{i\in T} (\hat{\rho}_i - \rho_i)(1-s_i) \right| \leq \sum_{i\in T} \epsilon(1-s_i) \leq k\epsilon,$$

where the last inequality holds because $0 \leq 1 - s_i \leq 1$. This implies

$$U(T) + k\epsilon \geq \hat{U}(T) \geq U(T) - k\epsilon. \tag{4}$$

Since we select $\hat{T}$ to maximize $\hat{U}(T)$, it satisfies

$$\hat{U}(\hat{T}) \geq \hat{U}(T^*) \geq U(T^*) - k\epsilon, \tag{5}$$

Combining (4) and (5), we bound $U(\hat{T})$ as:

$$U(\hat{T}) \geq \hat{U}(\hat{T}) - k\epsilon \geq (U(T^*) - k\epsilon) - k\epsilon = U(T^*) - 2k\epsilon.$$

Thus, $\sum_{i \in \hat{T}} \rho_i(1 - s_i) \geq \sum_{i \in T^*} \rho_i(1 - s_i) - 2k\epsilon$, as required.

## A.7 Proof of Lemma 4

We demonstrate that the invariant holds by using induction. First, we verify that before any computation has begun, the invariant is satisfied by the initialized values:

$$\boldsymbol{\Delta} - \hat{\boldsymbol{\Delta}}^{(0)} = \boldsymbol{\Delta} - \boldsymbol{0} = (\boldsymbol{1} - \boldsymbol{s}) \odot \boldsymbol{M}^\top \boldsymbol{1}.$$

Let $\boldsymbol{r}_a^{(t)}$ denote the residual vector before an update task for any node $i$, and let $\boldsymbol{r}_a^{(t+1)}$ denote the residual vector after the update task. Then a single update task corresponds to the following steps:

$$\hat{\boldsymbol{\Delta}}^{(t+1)} = \hat{\boldsymbol{\Delta}}^{(t)} + \boldsymbol{r}_a^{(t)}(i)(\boldsymbol{1} - \boldsymbol{s}) \odot \boldsymbol{R}\boldsymbol{e}_i,$$
$$\boldsymbol{r}_a^{(t+1)} = \boldsymbol{r}_a^{(t)} - \boldsymbol{r}_a^{(t)}(i)\boldsymbol{e}_i + \boldsymbol{r}_a^{(t)}(i)\boldsymbol{P}^\top(\boldsymbol{I} - \boldsymbol{R})\boldsymbol{e}_i = \boldsymbol{r}_a^{(t)} - \boldsymbol{r}_a^{(t)}(i)(\boldsymbol{I} - \boldsymbol{P}^\top(\boldsymbol{I} - \boldsymbol{R}))\boldsymbol{e}_i$$

Assuming that $\boldsymbol{\Delta} - \hat{\boldsymbol{\Delta}}^{(t)} = (\boldsymbol{1} - \boldsymbol{s}) \odot \boldsymbol{M}^\top \boldsymbol{r}_a^{(t)}$, it follows that

$$\begin{aligned}
\boldsymbol{\Delta} - \hat{\boldsymbol{\Delta}}^{(t+1)} &= \boldsymbol{\Delta} - \hat{\boldsymbol{\Delta}}^{(t)} - \boldsymbol{e}_i^\top(\boldsymbol{1} - \boldsymbol{s}) \odot \boldsymbol{R}\boldsymbol{r}_a^{(t)} \\
&= (\boldsymbol{1} - \boldsymbol{s}) \odot \boldsymbol{M}^\top \boldsymbol{r}_a^{(t)} - \boldsymbol{r}_a^{(t)}(i)(\boldsymbol{1} - \boldsymbol{s}) \odot \boldsymbol{R}\boldsymbol{e}_i \\
&= (\boldsymbol{1} - \boldsymbol{s}) \odot \boldsymbol{M}^\top \left( \boldsymbol{r}_a^{(t+1)} + \boldsymbol{r}^{(t)}(i)(\boldsymbol{I} - \boldsymbol{P}^\top(\boldsymbol{I} - \boldsymbol{R}))\boldsymbol{e}_i \right) - \boldsymbol{r}_a^{(t)}(i)(\boldsymbol{1} - \boldsymbol{s}) \odot \boldsymbol{R}\boldsymbol{e}_i \\
&= (\boldsymbol{1} - \boldsymbol{s}) \odot \boldsymbol{M}^\top \boldsymbol{r}_a^{(t+1)},
\end{aligned}$$

which completes the proof.

## A.8 Proof of Lemma 5

On the one hand, since $\boldsymbol{r}_a \geq \boldsymbol{0}$, by Lemma 4, we have $\boldsymbol{\Delta}(v) \geq \hat{\boldsymbol{\Delta}}(v), \forall v \in V$; on the other hand, the condition in Line 2 of Algorithm 1 indicates that, after termination, the vector $\boldsymbol{r}_a \leq \epsilon\boldsymbol{1}$, then we have that:

$$\boldsymbol{\Delta} - \hat{\boldsymbol{\Delta}} = (\boldsymbol{1} - \boldsymbol{s}) \odot \boldsymbol{M}^\top \boldsymbol{r}_a \leq \epsilon(\boldsymbol{1} - \boldsymbol{s}) \odot \boldsymbol{M}^\top \boldsymbol{1} = \epsilon\boldsymbol{\Delta}. \tag{6}$$

The equation above implies that $(1 - \epsilon)\boldsymbol{\Delta}(v) \leq \hat{\boldsymbol{\Delta}}(v), \forall v \in V$, hence the estimator $\hat{\boldsymbol{\Delta}}$ returned by Algorithm 1 satisfies $(1 - \epsilon)\boldsymbol{\Delta}(v) \leq \hat{\boldsymbol{\Delta}}(v) \leq \boldsymbol{\Delta}(v), \forall v \in V$.

## A.9 Proof of Lemma 6

We demonstrate that the invariant holds by using induction. First, we verify that before any computation has begun, the invariant is satisfied by the initialized values for any node $v \in V$:

$$\boldsymbol{\Delta}(v) - (\hat{\boldsymbol{\Delta}}(v) + \tilde{\Delta}) = \boldsymbol{\Delta}(v) - \hat{\boldsymbol{\Delta}}(v) = (1 - \boldsymbol{s}(v)) \cdot \boldsymbol{e}_v^\top \boldsymbol{M}^\top \boldsymbol{r}_a = \boldsymbol{r}_a^\top \boldsymbol{M}\boldsymbol{R}^{-1}\boldsymbol{r}^0.$$

Let $\boldsymbol{r}_s^{(t)}$ denote the residual vector before an update task for any node $u$, and let $\boldsymbol{r}_s^{(t+1)}$ denote the residual vector after the update task. Then a single update task corresponds to the following steps:

$$\tilde{\Delta}^{(t+1)} = \tilde{\Delta}^{(t)} + \boldsymbol{r}_a(u) \cdot \boldsymbol{r}_s^{(t)}(u),$$
$$\boldsymbol{r}_s^{(t+1)} = \boldsymbol{r}_s^{(t)} - \boldsymbol{r}_s^{(t)}(u)\boldsymbol{e}_u + \boldsymbol{r}_s^{(t)}(u)(\boldsymbol{I} - \boldsymbol{R})\boldsymbol{P}\boldsymbol{e}_u = \boldsymbol{r}_s^{(t)} - \boldsymbol{r}_s^{(t)}(u)(\boldsymbol{I} - (\boldsymbol{I} - \boldsymbol{R})\boldsymbol{P})\boldsymbol{e}_u$$

Assuming that $\boldsymbol{\Delta}(v) - (\hat{\boldsymbol{\Delta}}(v) + \tilde{\Delta}^{(t)}) = \boldsymbol{r}_a^\top \boldsymbol{M}\boldsymbol{R}^{-1}\boldsymbol{r}_s^{(t)}$, it follows that

$$\begin{aligned}
\boldsymbol{\Delta}(v) - (\hat{\boldsymbol{\Delta}}(v) + \tilde{\Delta}^{(t+1)}) &= \boldsymbol{\Delta}(v) - (\hat{\boldsymbol{\Delta}}(v) + \tilde{\Delta}^{(t)} + \boldsymbol{r}_a(u) \cdot \boldsymbol{r}_s^{(t)}(u)) \\
&= \boldsymbol{r}_a^\top \boldsymbol{M}\boldsymbol{R}^{-1}\boldsymbol{r}_s^{(t)} - \boldsymbol{r}_a(u) \cdot \boldsymbol{r}_s^{(t)}(u) \\
&= \boldsymbol{r}_a^\top \boldsymbol{M}\boldsymbol{R}^{-1} \left( \boldsymbol{r}_s^{(t+1)} + \boldsymbol{r}_s^{(t)}(u)(\boldsymbol{I} - (\boldsymbol{I} - \boldsymbol{R})\boldsymbol{P})\boldsymbol{e}_u \right) - \boldsymbol{r}_a(u) \cdot \boldsymbol{r}_s^{(t)}(u) \\
&= \boldsymbol{r}_a^\top \boldsymbol{M}\boldsymbol{R}^{-1}\boldsymbol{r}_a^{(t+1)},
\end{aligned}$$

which completes the proof.

## A.10 Proof of Lemma 7

On the one hand, since $\boldsymbol{r}_s \geq \boldsymbol{0}$, by Lemma 6, we have $\boldsymbol{\Delta}(v) - (\hat{\boldsymbol{\Delta}}(v) + \tilde{\Delta}) \geq 0$; on the other hand, the condition in Line 2 of Algorithm 2 indicates that, after termination, the vector $\boldsymbol{r}_s \leq \epsilon \boldsymbol{R1}$, then we have that

$$\boldsymbol{\Delta}(v) - (\hat{\boldsymbol{\Delta}}(v) + \tilde{\Delta}) = \boldsymbol{r}_a^\top \boldsymbol{M} \boldsymbol{R}^{-1} \boldsymbol{r}_a \leq \epsilon \boldsymbol{r}_a^\top \boldsymbol{M1} = \epsilon \cdot (\boldsymbol{r}_a)_{\text{sum}}.$$

Hence the estimator $\tilde{\Delta}$ returned by Algorithm 2 satisfies $0 < \boldsymbol{\Delta}(v) - (\hat{\boldsymbol{\Delta}}(v) + \tilde{\Delta}) \leq \epsilon \cdot (\boldsymbol{r}_a)_{\text{sum}}$.

## A.11 Proof of Lemma 8

We present the proof for absolute errors; the relative error case follows analogously through error bound transformations and is thus omitted.

Let $\mathcal{S}^*$ denote the optimal size-$k$ node set. We first show that any node $i \in T$ must belong to $\mathcal{S}^*$. By definition of $T$, we have $\hat{\boldsymbol{a}}(i) \geq \hat{\boldsymbol{a}}_{\langle k+1 \rangle} + \epsilon$. Applying the error bound yields $\boldsymbol{a}(i) \geq \hat{\boldsymbol{a}}(i) \geq \hat{\boldsymbol{a}}_{\langle k+1 \rangle} + \epsilon$. For any node $j \notin \mathcal{S}^*$, it holds that

$$\boldsymbol{a}(j) \leq \boldsymbol{a}_{\langle k+1 \rangle} \leq \hat{\boldsymbol{a}}_{\langle k+1 \rangle} + \epsilon \leq \boldsymbol{a}(i).$$

This shows $\boldsymbol{a}(i) \geq \boldsymbol{a}(j)$ for all $j \notin \mathcal{S}^*$, which implies $i \in \mathcal{S}^*$.

Next, we prove that $\mathcal{S}^* \setminus T \subseteq C$. Suppose for contradiction that there exists $i \in \mathcal{S}^*$ with $i \notin T \cup C$. By definition of $C$, this requires $\hat{\boldsymbol{a}}(i) < \hat{\boldsymbol{a}}_{\langle k \rangle} - \epsilon$. Using the error bound, we obtain

$$\boldsymbol{a}(i) \leq \hat{\boldsymbol{a}}(i) + \epsilon < \hat{\boldsymbol{a}}_{\langle k \rangle}.$$

However, since $i \in \mathcal{S}^*$, optimality implies $\boldsymbol{a}(i) \geq \boldsymbol{a}_{\langle k \rangle} \geq \hat{\boldsymbol{a}}_{\langle k \rangle}$, which contradicts the above inequality. Therefore, $\mathcal{S}^* \setminus T \subseteq C$ must hold.

## A.12 Proof of Theorem 3

We first proof Lemma 11, which explains the upper bound of the time complexity of Algorithm 1.

**Lemma 11.** *An upper bound on the running time of Algorithm 1 is* $O(\frac{d_{\max}}{\alpha_{\min}} n \log \frac{1}{n})$.

*Proof.* During the processing, we add a dummy node that does not actually exist. This node is initially placed at the head of the queue and is re-appended to the queue each time it is popped. The set of nodes in the queue when this dummy node is processed for the $(i + 1)$-th time is regarded as $S^{(i)}$, and the residue vector at this time is regarded as $\boldsymbol{r}_a^{(i)}$. The sum of the vector $\boldsymbol{r}_a^{(i)}$ is denoted as $(\boldsymbol{r}_a)_{\text{sum}}^{(i)}$. The process of handling this set is considered the $(i + 1)$-th iteration. In the context of the $(i + 1)$-th iteration, when node $v \in S^{(i)}$ is about to be processed, it holds that $\boldsymbol{r}_a(v) \geq \boldsymbol{r}_a^{(i)}(v)$. Upon the completion of this operation, the sum of residual vector is decreased by $\alpha_v \boldsymbol{r}_a(v)$. Consequently, by the conclusion of the $(i + 1)$-th iteration, the total reduction in the sum of residue vector amounts to:

$$(\boldsymbol{r}_a)_{\text{sum}}^{(i)} - (\boldsymbol{r}_a)_{\text{sum}}^{(i+1)} = \sum_{v \in S^{(i)}} \alpha_v \boldsymbol{r}_a(v) \geq \sum_{v \in S^{(i)}} \alpha_v \boldsymbol{r}_a^{(i)}(v). \tag{7}$$

Given that the bound for any node $v$ is $\epsilon$, we obtain:

$$\frac{\sum_{v \in S^{(i)}} \boldsymbol{r}_a^{(i)}(v)}{|S^{(i)}|} \geq \epsilon \quad \text{and} \quad \frac{\sum_{v \notin S^{(i)}} \boldsymbol{r}_a^{(i)}(v)}{|V \setminus S^{(i)}|} \leq \epsilon.$$

Therefore, it follows that:

$$\frac{\sum_{v \in S^{(i)}} \boldsymbol{r}_a^{(i)}(v)}{|S^{(i)}|} \geq \frac{\sum_{v \in S^{(i)}} \boldsymbol{r}_a^{(i)}(v) + \sum_{v \notin S^{(i)}} \boldsymbol{r}_a^{(i)}(v)}{|S^{(i)}| + |V \setminus S^{(i)}|} = \frac{(\boldsymbol{r}_a)_{\text{sum}}^{(i)}}{n}.$$

Substituting the above expression into Eq. (7), we obtain

$$(\boldsymbol{r}_a)_{\text{sum}}^{(i+1)} \le (\boldsymbol{r}_a)_{\text{sum}}^{(i)} - \sum_{v \in S^{(i)}} \alpha_v \boldsymbol{r}_a^{(i)}(v) \le (\boldsymbol{r}_a)_{\text{sum}}^{(i)} - \alpha_{\min} \sum_{v \in S^{(i)}} \boldsymbol{r}_a^{(i)}(v)$$

$$\le (1 - \frac{\alpha_{\min}}{n}|S^{(i)}|)(\boldsymbol{r}_a)_{\text{sum}}^{(i)}$$

$$\le \prod_{t=0}^{i}(1 - \frac{\alpha_{\min}}{n}|S^{(t)}|)(\boldsymbol{r}_a)_{\text{sum}}^{(0)}.$$

By utilizing the fact that $1 - x \le e^{-x}$, we obtain

$$(\boldsymbol{r}_a)_{\text{sum}}^{(i+1)} \le \exp\left(-\frac{\alpha_{\min}}{n}(\sum_{t=0}^{i}|S^{(t)}|)\right)n. \tag{8}$$

Let $T^{(i+1)} = \sum_{t=0}^{i}|S^{(t)}|$ be defined as the total number of updates before the start of the $(i+1)$-th iteration. According to Eq. (8), to satisfy $(\boldsymbol{r}_a)_{\text{sum}}^{(i+1)} \le \epsilon n$, it suffices to find the minimum number of updates that meets the following conditions:

$$\exp\left(-\frac{\alpha_{\min}}{n}T^{(i+1)}\right) \le \epsilon \le \exp\left(-\frac{\alpha_{\min}}{n}T^{(i)}\right).$$

Thus, we obtain $T^{(i)} \le \frac{1}{\alpha_{\min}}n\log\frac{1}{\epsilon} \le T^{(i+1)}$, Given the fact that $T^{(i+1)} - T^{(i)} = |S^{(i)}| \le n$, we further derive

$$T^{(i+1)} \le T^{(i)} + n \le \frac{1}{\alpha_{\min}}n\log\frac{1}{\epsilon} + n.$$

For node $v$, the push operation reduces $(\boldsymbol{r}_a)_{\text{sum}}$ by $\alpha_v \boldsymbol{r}_a(v)$. Consequently, after incurring a total number of updates $T$, the reduction in $(\boldsymbol{r}_a)_{\text{sum}}$ is at least $\epsilon\alpha_v T$. Hence starting from the state of $(\boldsymbol{r}_a)_{\text{sum}} \le \epsilon n$, the time cost $T$ is bounded by $O(n/\alpha_{\min})$, thereby constraining the overall time complexity of Algorithm 1 to $O(\frac{d_{\max}^{+}}{\alpha_{\min}}n\log\frac{1}{\epsilon})$. $\qquad\square$

For the refinement stage, we first proof the time complexity of Line 7 in Algorithm 3, when the absolute error parameter $\epsilon = 1$. Assuming that estimating node is $v \in V$, the contribution of $\boldsymbol{r}_s$ at node $u \in V$ each time is $\boldsymbol{r}_s(u)$. Due to the existence of boundary $\boldsymbol{h} = \frac{1}{(\boldsymbol{r}_a)_{\text{sum}}}\boldsymbol{R}\mathbf{1}$, its minimum value is $\alpha_u$, and the upper bound on the total contribution is $(1 - \boldsymbol{s}(v))\boldsymbol{M}(u, v)$. Therefore, the upper bound on the number of updates at node $u$ is $\frac{(\boldsymbol{r}_a)_{\text{sum}}\cdot(1-\boldsymbol{s}(v))\boldsymbol{M}(u,v)}{\alpha_u}$. Therefore, assuming that the candidate set returned in Line 3 of Algorithm 3 is $C_a$, the upper bound of the time complexity during the first round of execution in the refinement stage is

$$\sum_{v \in C_a}\sum_{u \in V}\frac{(\boldsymbol{r}_a)_{\text{sum}} \cdot (1 - \boldsymbol{s}(v))\boldsymbol{M}(u, v)d_u^{-}}{\alpha_u} \le \frac{(\boldsymbol{r}_a)_{\text{sum}} \cdot d_{\max}^{-}}{\alpha_{\min}}\sum_{u \in V}\sum_{v \in C_a}\boldsymbol{M}(u, v)$$

$$= \frac{(\boldsymbol{r}_a)_{\text{sum}} \cdot d_{\max}^{-}}{\alpha_{\min}} \cdot \sum_{v \in C_a}\rho_v \le \frac{\epsilon' \cdot d_{\max}^{-}n}{\alpha_{\min}} \cdot \sum_{v \in C_a}\rho_v.$$

Now we consider when absolute error parameter $\epsilon = \frac{1}{2}, \frac{1}{4}, \ldots, \frac{1}{2n}, \ldots$. We assume that the error in one round of the process is $\epsilon'$, so the error in the previous round of the process is $2\epsilon'$. The upper bound of time complexity of this round is

$$\sum_{v \in C_a}\sum_{u \in V}\frac{2\epsilon'/(\boldsymbol{r}_a)_{\text{sum}} \cdot (1 - \boldsymbol{s}(v))(\boldsymbol{M}\boldsymbol{R}^{-1}\boldsymbol{R}\mathbf{1})(u)d_u^{-}}{\epsilon'/(\boldsymbol{r}_a)_{\text{sum}}\alpha_u} \le \frac{2}{\alpha_{\min}}\sum_{v \in C_a}\sum_{u \in V}(\boldsymbol{M}\mathbf{1})(u)d_u^{-} = |C_a| \cdot \frac{2m}{\alpha_{\min}}$$

When the current error $\epsilon'$ is $\frac{1}{2i}$, the upper bound of the total time complexity from $\epsilon' = \frac{1}{2}$ to $\epsilon' = \frac{1}{2i}$ is $|C_a| \cdot \frac{2i \cdot m}{\alpha_{\min}}$. Hence, when $\epsilon' < k_{\text{gap}}$, the upper bound of the total time complexity is $|C_a| \cdot \frac{m}{k_{\text{gap}} \cdot \alpha_{\min}}$.

Hence, the upper bound of time complexity of Algorithm 3 is

$$O(\frac{d_{\max}^+ n}{\alpha_{\min}}(\log \frac{1}{\epsilon} + \epsilon \sum_{v \in C_a} \rho_v) + \frac{|C_a|m}{k_{\text{gap}} \cdot \alpha_{\min}}).$$

Under normal circumstances, when $\epsilon$ is sufficiently small, we obtain $|C_a|$ as either 0 or a relatively small constant. Therefore, we can derive the upper bound of time complexity as $O(\frac{d_{\max}^+ n}{\alpha_{\min}} \log \frac{1}{\epsilon} + \frac{m}{k_{\text{gap}} \cdot \alpha_{\min}})$ in the general case.

## B  Forest Sampling Algorithm

For a graph $\mathcal{G} = (V, E)$, a *spanning subgraph* of $\mathcal{G}$ is a subgraph with node set $V$ and edge set being a subset of $E$. A *converging tree* is a weakly connected graph where exactly one node, called the *root*, has out-degree 0, while all other nodes have out-degree 1. An isolated node is considered a trivial converging tree with itself as the root. A *spanning converging forest* of $\mathcal{G}$ is a spanning subgraph in which every weakly connected component is a converging tree. This structure coincides with the notion of an *in-forest* as introduced by [50] and further studied by [51]. Spanning converging forests are closely related to the fundamental matrix of our model. Following the Lemma 3 and its Proof A.4, it can be shown that the any entry of the fundamental matrix can be represented in the form of spanning converging forest.

Wilson's algorithm [52] provides an efficient method for generating uniform spanning trees by leveraging loop-erased absorbing random walks. For any graph $\mathcal{G} = (V, E)$, Wilson's algorithm can be adapted to generate a uniform spanning converging forest, by using the method similar to that in [53, 30]. The details of the algorithm are presented in 4.

---

**Algorithm 4:** RANDOMFOREST($\mathcal{G}$, $\boldsymbol{R}$)

---

**Input**   : graph $\mathcal{G}$, resistance matrix $\boldsymbol{R}$.
**Output** : root index vector RootIndex.

1 **for** $i \leftarrow 1$ **to** $n$ **do**
2   | InForest[$i$] $\leftarrow$ false; Next[$i$] $\leftarrow$ -1; RootIndex[$i$] $\leftarrow$ 0;

3 **for** $i \leftarrow 1$ **to** $n$ **do**
4   | $u \leftarrow i$;
5   | **while** *not InForest[u]* **do**
6   |   | **if** RAND() $\leq \alpha_u$ **then**
7   |   |   | InForest[$u$] $\leftarrow$ true
8   |   |   | Next[$u$] $\leftarrow -1$
9   |   |   | RootIndex[$u$] $\leftarrow u$
10  |   | **else**
11  |   |   | $u \leftarrow$ Next[$u$] $\leftarrow$ RANDOMSUCCESSOR($u$, $\mathcal{G}$)
12  | RootNow $\leftarrow$ RootIndex[$u$]
13  | **for** $u \leftarrow i$; *not InForest[u]*; $u \leftarrow$ *Next[u]* **do**
14  |   | InForest[$u$] $\leftarrow$ true
15  |   | RootIndex[$u$] $\leftarrow$ RootNow

16 **return** RootIndex

---

The Algorithm 4 initializes three vectors: *InForest* (marks nodes added to the spanning forest), *Next* (tracks random walk steps), and *RootIndex* (records root assignments), all set to false, -1, and 0 respectively (Line 1). For each node in order (Line 3), it performs a loop-erased absorbing random walk that terminates either: (1) with probability $\alpha_u$ (making $u$ a new root, Line 6-7), or (2) upon hitting existing forest nodes (Line 5). The walk's path is recorded in *Next*. After termination, the algorithm backtracks along the walk path (Line 13-14), adding all nodes to the forest and assigning them the current root index. This continues until all nodes are processed (Line 3), returning the *RootIndex* vector representing the sampled rooted spanning forest (Line 15). Then, based on Lemma 3, we obtained the implementation of our algorithm FOREST, and the pseudocode is shown in Algorithm 5.

**Algorithm 5:** Forest($\mathcal{G}$, $\boldsymbol{R}$, $\boldsymbol{s}$, $l$)

**Input** : graph $\mathcal{G}$, resistance matrix $\boldsymbol{R}$, internal opinion vector $\boldsymbol{s}$, sample count $l$.

**Output** : the target set $\hat{T}$.

**1 Initialize** : $\hat{T} \leftarrow \emptyset$; $\hat{\boldsymbol{\rho}} \leftarrow \boldsymbol{0}$

**2 for** $t \leftarrow 1$ **to** $l$ **do**

**3**     RootIndex $\leftarrow$ RANDOMFOREST($\mathcal{G}$, $\boldsymbol{R}$)

**4**     **for** $i \leftarrow 1$ **to** $n$ **do**

**5**        $u \leftarrow$ RootIndex$[i]$

**6**        $\hat{\boldsymbol{\rho}}_u \leftarrow \hat{\boldsymbol{\rho}}_u + 1$

**7** $\hat{\boldsymbol{\rho}} \leftarrow \hat{\boldsymbol{\rho}}/l$

**8 for** $i = 1$ **to** $n$ **do**

**9**     $u \leftarrow \arg\max_{v \in V \setminus \hat{T}} \hat{\boldsymbol{\rho}}_v (1 - \boldsymbol{s}_v)$

**10**     $\hat{T} \leftarrow \hat{T} \cup \{u\}$

**11 return** $\hat{T}$

## C    Additional Experiments

### C.1    Efficiency

Figure 3 and Table 3 collectively present a comprehensive performance analysis of our proposed MIS algorithm under different parameters and network conditions. Figure 3 demonstrates the algorithm's runtime performance across all tested networks under various resistance coefficient distributions as parameter k varies. The results indicate that although fluctuations in the potential influence distribution near the k-th most influential node cause minor variations, MIS consistently maintains superior performance compared to both RWB and FOREST algorithms in all test cases.

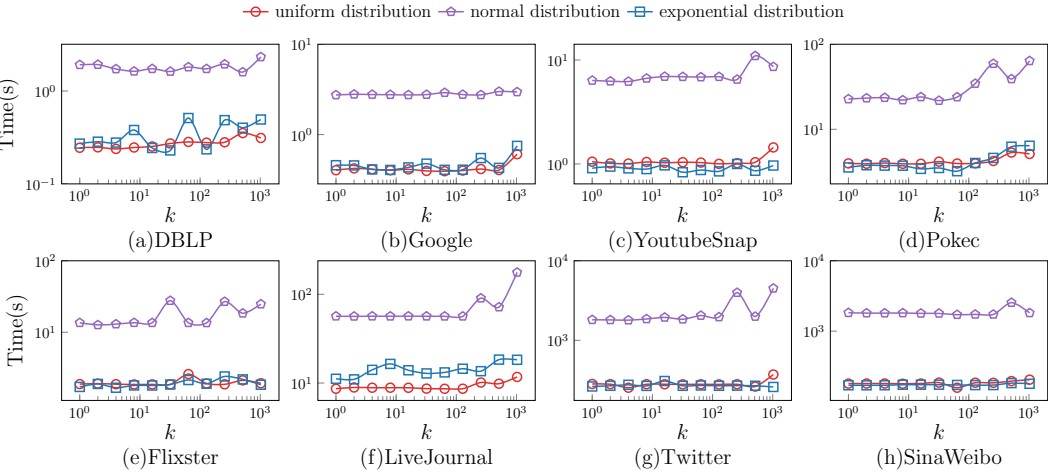

Figure 3: Running time of algorithm MIS under different resistance coefficient distributions.

Complementing the runtime analysis, Table 3 evaluates computational efficiency by examining the average number of updates per node during algorithm convergence under the specific parameter setting of k=64 and uniform resistance coefficients. The update ratios for all networks remain within a practical range of 27.06 to 249.28, with Twitter maintaining controllable levels despite its largest scale while Google achieves optimal efficiency. These results confirm that the algorithm achieves near-linear time complexity in practice, with stable constant factors that do not increase significantly with network size, demonstrating strong scalability for large-scale network applications.

Table 3: Updates per node across networks.

| Network | DBLP | Google | YoutubeSnap | Pokec | Flixster | LiveJournal | Twitter | SinaWeibo |
|---------|------|--------|-------------|-------|----------|-------------|---------|-----------|
| Ratio | 49.93 | 27.06 | 49.34 | 126.86 | 114.84 | 106.05 | 249.28 | 78.77 |

## C.2 Accuracy

Extending the analysis from Figures 1 and 2, we conduct a comprehensive evaluation of accuracy performance across multiple experimental configurations. Figures 4 and 5 examine performance under normal distribution conditions. Similarly, Figures 6 and 7 demonstrate algorithm behavior under exponential distribution. The complete set of experimental results reveals several important findings. First, our proposed algorithms (RWB, FOREST, and MIS) consistently outperform all baseline methods across every tested condition. While the RWB algorithm fails to complete execution within practical time limits for certain cases due to its computational complexity, the algorithms FOREST and MIS successfully deliver results in all scenarios. More significantly, the MIS algorithm achieves exact solutions in every case while simultaneously maintaining near-perfect accuracy (NDCG $\approx 1$) in sequence ordering. This combination of guaranteed precision and optimal ordering performance clearly establishes the superiority of our MIS approach over both baseline methods and our own alternative proposals.

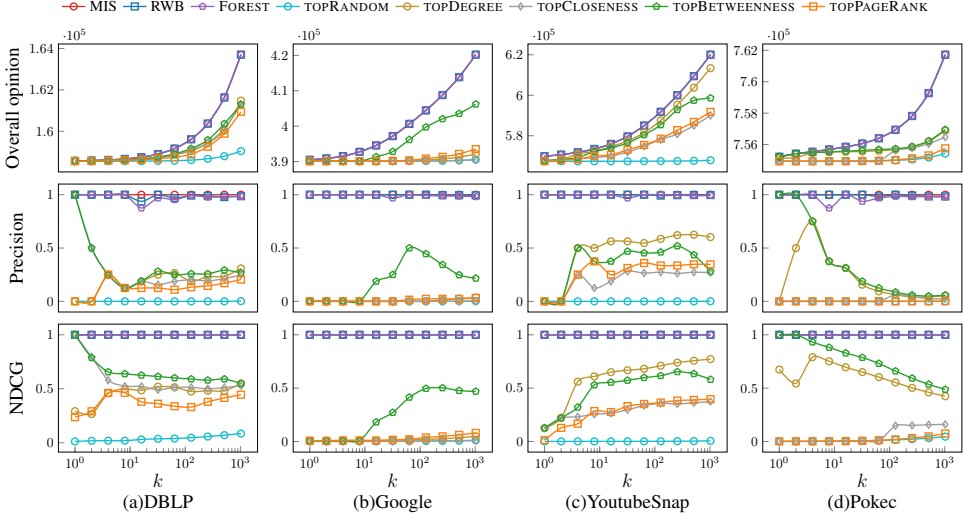

Figure 4: Performance of algorithms in accuracy under normal distribution.

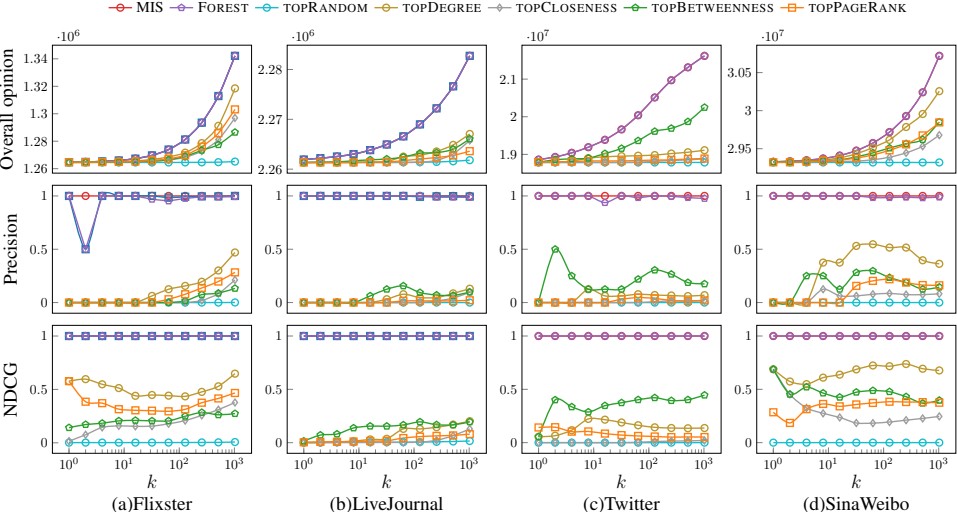

Figure 5: Performance of algorithms in accuracy under normal distribution.

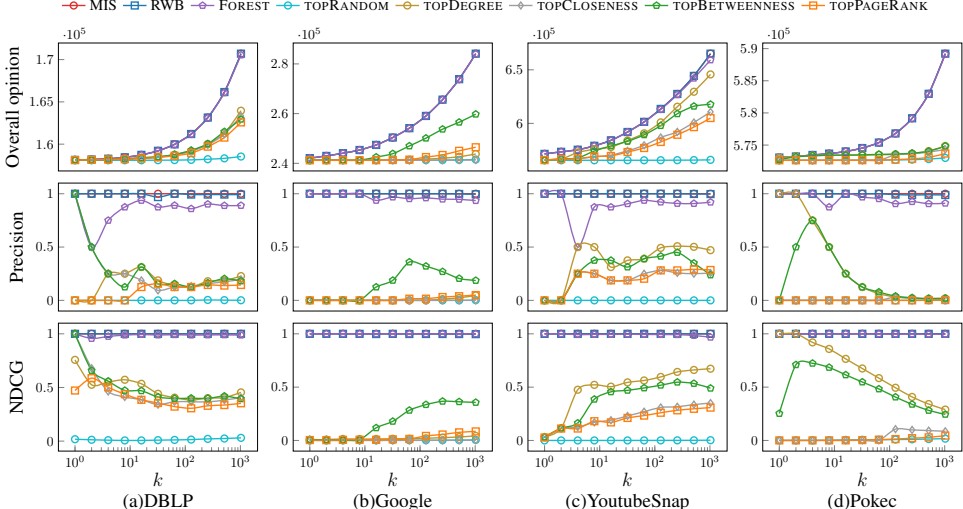

Figure 6: Performance of algorithms in accuracy under exponential distribution.

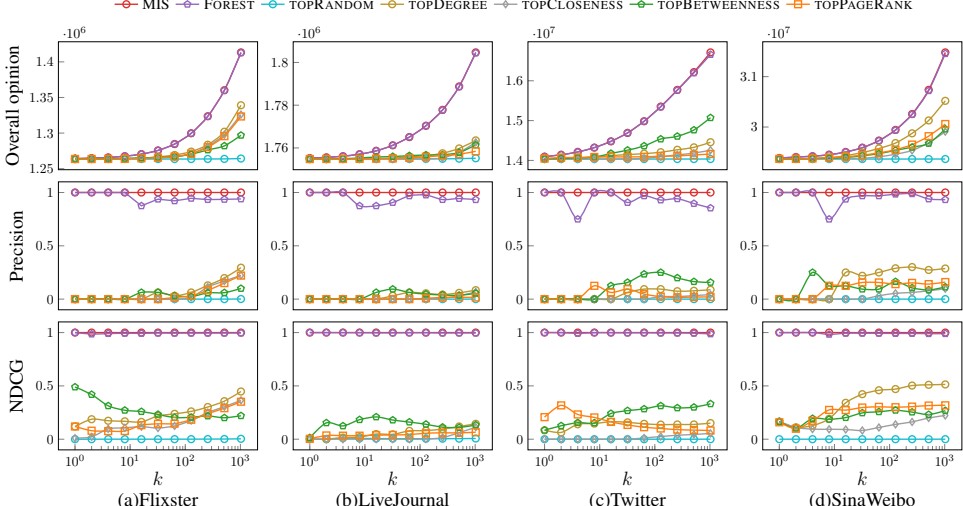

Figure 7: Performance of algorithms in accuracy under exponential distribution.

# D  Dataset Details

The study utilizes eight benchmark datasets obtained from the Koblenz Network Collection [45], SNAP [46], and Network Repository [47].

DBLP is an undirected co-authorship network where nodes represent authors and edges indicate co-authorship relationships. The dataset uses the largest connected component, with publication venues (conferences or journals) defining ground-truth communities, retaining the top 5,000 high-quality communities each containing at least three nodes. Google is a directed network where nodes represent web pages and edges represent hyperlinks, originating from the 2002 Google Programming Contest. YouTube is an undirected social network where nodes represent users and edges represent friendships, with communities defined by user-created groups, similarly retaining the top 5,000 communities with at least three nodes and using the largest connected component. Pokec is a Slovak social network where nodes represent users and directed edges represent friendships, containing anonymized user attributes such as gender and age. Flixster is an undirected movie social network where nodes represent users and edges represent social connections. LiveJournal is a directed online community network where nodes represent users and edges represent friend relationships. Twitter is a directed social network where nodes represent users and edges represent follower relationships. SinaWeibo is a directed microblogging social network where nodes represent users and edges represent social connections. Detailed statistical characteristics of each network are provided in Table 1.

