# OpenReview forum: "Opinion Maximization in Social Networks by Modifying Internal Opinions"
_NeurIPS.cc/2025/Conference — NeurIPS 2025 poster_

### Official Review · Reviewer_2WFV · 2025-06-30

**Clarity:** 3
**Significance:** 3
**Originality:** 3
**Rating:** 5
**Confidence:** 3

**Summary:**

The paper deals with the problem of strategically governing the public opinion formation process in a social network in order to maximize the overall (average) opinion expressed by the agents. This is a fundamental problem in the opinion dynamics literature, and it has extensively studied under the lens of both social science and computer science lens.
In the literature several methods have been proposed to influence the opinion formation process, depending on the adopted opinion dynamics model. In this paper, a variation of the Friedkin-Jonsen opinion dynamics model is adopted, where each agent has a (fixed) internal opinion and an (variable) expressed opinion and at each step they decide to express an opinion that depends on their own internal opinion and the opinions expressed by their neighbors in the previous step. The technique used to influence the public opinion is to change the internal opinions of a selected set of nodes.

The paper addresses the optimization problem of determining a set of k nodes and have their internal opinions modified in order to maximize the overall opinion expressed at the equilibrium of the opinion dynamics. The paper focus is on the computational complexity and some efficient algorithms are proposed. The main result of the paper is a deterministic asynchronous algorithm that exactly identifies the optimal set of nodes through asynchronous update operations and progressive refinement, ensuring both efficiency and precision.

An extensive set of experiments on real-world datasets are presented to validate the efficiency of the proposed algorithms with respect to baseline approaches and show their scalability even on
networks with tens of millions of nodes.

**Questions:**

1.	Lines 80-81 and line 99: usually in opinion dynamics models such as DeGroot and FJ the graph is weighted, where weights represent the strength of the influence of an agent on a neighbour. Why do you adopt a non-weighted scenario? What does represent an edge in you model? Can your results hold even in the weighted case?
2.	Lines 119-120: What is the meaning of M(u, v)? Is the influence of v on the expressed opinion of u?
3.	Line 138: As a matter of fact your optimization problem is to select a set T of k nodes that maximized the increment in the overall opinion. Thus, f_U(R, s) = f(R, s) + \sum_{u\in U} \Delta(u) and you are looking for \arg\max_U f_U(R, s).

Other comments/suggestions:

1.	Lines 19-21, some references could be useful to support your statement of the importance of the overall opinion metric.
2.	Lines 80-86: even if you do not specify if the graph is directed or undirected all your definitions are given for directed graphs (you are distinguishing between in-degree and out-degree). It could be better specify that definitions are given for directed graphs and can be easily extended to the undirected case.
3.	Line 113: I suggest to write f(R, s) = \sum_v s_v \sum_u M(u, v). Moreover, I would maintain for all the paper the notation of using u, v to denote nodes and use i only as index. You should also maintain the same notation for all the appendices.
4.	Line 163: a definition of the absorption probability vector of absorbing random walks starting at node i ∈ V could be useful.
5.	Line 190: as for the previous comment, a definition of a spanning converging forest could be helpful.
6.	Line 235: I guess something is missing in this sentence.
7.	Line 240: Lemma 8 is stated only in the following.
8.	Line 323: A more informative caption for figure 1 could be helpful.

**Ethical Concerns:**

["NO or VERY MINOR ethics concerns only"]

**Final Justification:**

I read other reviews and authors' rebuttal. I agree with issues related to connections of the paper to the existing literature but I think that authors answered satisfactorily and the final version of the paper could fix all these points.
I confirm my positive feeling on the paper.

**Limitations:**

Yes.

**Paper Formatting Concerns:**

No major formatting issue has been observed.

**Quality:**

3

**Strengths And Weaknesses:**

STRENGHTS
The paper studies an important problem in the opinion dynamics literature, with significant practical applications. The argument is relevant and perfectly fits the scopus of the conference.
This is a theoretical solid paper, with interesting and original contributions. The paper also presents an extensive experimental analysis to validate efficiency, precision and scalability of the proposed algorithms on real-world datasets.
The organization of the paper is good and the presentation is sufficiently clear.

WEAKNESSES
A more careful reading of the paper would be necessary to fix several typos and solve inconsistencies in the notation, in particular between the main text and eppendices.
Appendices A and B are a simple list of proofs of lemmas and theorems. Some comments could be helpful for the reader.

---

> ### Author Rebuttal · Authors · 2025-07-30
>
> We sincerely appreciate your insightful feedback and the constructive suggestions provided.
>
> **W1: Writing Quality**
>
> We have carefully reviewed your comments and will meticulously implement the necessary revisions in the final submission to address all identified issues. Thank you.
>
> **Q1: Weighted Graph Adaptation**
>
> In our model, edges represent the direct connection strength between nodes. We focus on unweighted scenarios primarily because forest sampling algorithms face significant challenges in obtaining uniform samples from weighted graphs—this stems from the fact that forest weights are inherently tied to edge weights. To our knowledge, existing literature on forest sampling exclusively addresses unweighted graphs. Fortunately, our algorithm and its theoretical proofs can be seamlessly extended to weighted graphs, although no established baseline methods currently exist for weighted scenarios. For the sake of coherence and conciseness, we restrict our analysis to unweighted graphs in the main text.
>
> Regarding weighted graphs, let the edge weight from node $i$ to node $j$ be denoted as $w(i,j)$, and the out-degree of node $i$ be
> $d_i^+ = \sum_{ j \in N_{out}(i) } w(i,j)$. To adapt MIS for weighted graphs, the following modifications are required:
>
> - **Algorithm 1**: Replace the coefficient of $\boldsymbol{r}_a(v)$ in Line 5 from $\frac{1-\alpha_v}{d^+_v}$ to $\frac{(1-\alpha_v)w(v,u)}{d^+_v}$.
> - **Algorithm 2**: Replace the coefficient of $\boldsymbol{r}_s(v)$ in Line 5 from $\frac{1-\alpha_u}{d^+_u}$ to $\frac{(1-\alpha_u)w(u,v)}{d^+_u}$.
> - **Algorithm 3**: No modifications needed.
>
> All theoretical proofs for MIS remain structurally identical to those in the unweighted case, requiring only straightforward extensions to account for edge weights. We sincerely appreciate your suggestion. In the final version, we will further elaborate on the advantages of our algorithm in handling weighted graphs, along with additional experiments to verify its correctness and effectiveness.
>
> **Q2: M(u,v) Interpretation**
>
> For the equation $\boldsymbol{z}(u) = \sum_{v \in V} \boldsymbol{M}(u,v) \boldsymbol{s}(v)$, since each term $\boldsymbol{M}(u,v)$ is non-negative and satisfies $\sum_{v\in V}\boldsymbol{M}(u,v) = 1$, the $u$-th row of matrix $\boldsymbol{M}$ represents a convex combination of the elements in vector $\boldsymbol{s}$ that determine $\boldsymbol{z}(u)$. $\boldsymbol{M}(u,v)$ quantifies the influence weight of node $v$'s internal opinion on node $u$'s expressed opinion. Specifically, for two nodes with identical initial opinions, a larger $\boldsymbol{M}(u,v)$ value indicates a stronger influence of node $v$'s opinion on node $u$'s expressed opinion. We appreciate your question and will provide a more detailed explanation of this quantity in the revised manuscript.
>
> **Q3: Optimization Problem Restatement**
>
> You are absolutely right. We will provide a detailed explanation of this issue in the revised version. Thank you very much for your insightful feedback!

---

### Official Review · Reviewer_TSx5 · 2025-07-02

**Clarity:** 1
**Significance:** 3
**Originality:** 3
**Rating:** 4
**Confidence:** 3

**Summary:**

The paper investigates the overall opinion optimization in online social networks by strategically altering the internal opinions of $k$ selected nodes to maximize global consensus. Recognizing the computational intractability of exact methods ($O(n^3)$), the authors explore two sampling-based algorithms and propose a novel asynchronous update algorithm. The goal is to identify a subset of influential nodes whose opinion modification will maximize the collective sentiment across the network. The proposed methods are evaluated on large-scale real-world networks, demonstrating superior performance over existing baselines in terms of effectiveness, efficiency, and scalability.

**Questions:**

It is not clear why the sampling algorithms are required. I guess that they give fast algorithms for calculating final optimized opinion, but fails to provide the optimal nodes. If finding the optimal nodes fast is the purpose of the paper, why are the sampling algorithms proposed then? What is the role of the algorithms in the bigger picture? It is discussed neither in the intro, nor in the paper.

**Ethical Concerns:**

["NO or VERY MINOR ethics concerns only"]

**Final Justification:**

I want to retain my score.

**Limitations:**

Also I did not find any detail discussion on the limitations of this approach (the asynchronous update).

**Quality:**

2

**Strengths And Weaknesses:**

Strengths

1. The paper addresses a problem with significant consequences, that is, optimizing public opinion in the context of online social networks. This problem has many impactful applications like elections, marketing, and health campaigns.

2. Scalability of Proposed Algorithm: The asynchronous update algorithm shows scalability, offering a promising direction for handling massive graphs efficiently with exact results.

3. Sufficient Evaluation: The authors conduct extensive experiments on real-world networks, providing empirical evidence for the advantages of their algorithms over standard baselines.


Weaknesses

1. Unclear Role and Justification of Sampling Algorithms: The purpose and necessity of the sampling algorithms are not clearly articulated in the introduction. It's not evident why sampling is needed in addition to or instead of optimization techniques. The paper lacks a discussion on the specific limitations sampling tries to overcome, and how it fits into the broader solution strategy.

2. It is not clearly stated whether an edge from node $i$ to node $j$ implies that $i$ influences $j$ or vice versa. This is a crucial detail in opinion dynamics and should be explicitly defined to avoid confusion about how information flows in the model. Particularly, the equation following line 99, explaining opinion of a node seems unreasonable based on the direction of the edge.

3. Forest Sampling algorithm requires an inverse operation, but the complexity of this computation is not discussed, nor are the limitations or trade-offs. Moreover, the merits of sampling algorithms are not well-articulated, and the paper fails to explain why sampling cannot directly solve the $k$-node selection optimally—this needs clearer theoretical or empirical justification.

4. The paper is too mathematical and lacks intuitive motivations at places. People with insufficient domain knowledge would struggle to read this paper.

---

> ### Author Rebuttal · Authors · 2025-07-30
>
> Thank you for your time and efforts. The following is our response to your concerns.
>
> **W1: Role of Sampling Algorithms**
>
> We appreciate your insightful question. The sampling algorithms are included to ensure methodological completeness. Specifically:
>
> 1. The absorbing random walk-based approach provides a probabilistic interpretation of the problem, while also serving as conceptual inspiration for our deterministic algorithm—Algorithm 1 can indeed be viewed as a deterministic asynchronous push variant of absorbing random walk probabilities.
> 2. The forest sampling algorithm, an extension of [27], serves as a baseline since our model reduces to [27]'s formulation when fixing $\alpha_i = \frac{1}{1+d^+_i}$. This pre-existing method (developed prior to our MIS proposal) warrants inclusion to acknowledge foundational work.
>
> By employing both algorithms as baselines, we demonstrate the superior efficiency and accuracy of our deterministic asynchronous approach. We will further clarify their necessity and connections to MIS in the revised manuscript.
>
> **W2: Edge Directionality in Opinion Dynamics**
>
> In directed graphs, an edge from node $i$ to node $j$ signifies that node $i$ receives influence from node $j$—specifically, node $j$'s expressed opinion impacts node $i$'s expressed opinion. This causal relationship is well-established in directed graph-based opinion dynamics models [7, 27, 33].
>
> To illustrate this logic: consider social platforms like YouTube where a follower relationship (node $a$ follows node $b$) creates a directed edge $(a,b)$. Here, content (e.g., videos) published by $b$ becomes visible to $a$, demonstrating how $b$'s opinions influence $a$. Consequently, the right-hand side of the equation after Line 99 mathematically captures the aggregated influence of nodes followed by $i$ on $i$'s opinions. This aligns perfectly with the described social mechanism, validating the equation's rationale.
>
> We appreciate your suggestion and will provide additional clarifications  in the revised manuscript.
>
> **W3: Forest Sampling Algorithm**
>
> We appreciate your insightful question. The forest sampling algorithm efficiently approximates elements of the fundamental matrix's inverse, streamlining our solution process. However, it's important to clarify that the algorithm itself doesn't involve explicit matrix inversion operations. Regarding the matrix $\boldsymbol{Q} = \boldsymbol{D}(\boldsymbol{I}-\boldsymbol{R})^{-1}\boldsymbol{R}$ in Line 188: while its formulation appears to require matrix inversion, the special structure of the resistance matrix $\boldsymbol{R}$ (defined as a diagonal matrix in Line 102) renders $(\boldsymbol{I}-\boldsymbol{R})^{-1}$ computationally tractable without complex inversion procedures.
>
> Historically, before our deterministic asynchronous MIS algorithm, sampling methods provided practical advantages by mitigating time/space costs associated with high-complexity matrix inversions. Yet, as evidenced by Lemma 2, sampling only guarantees probabilistic error bounds (with probabilities <1), preventing exact problem-solving as formalized in Corollary 1. In contrast, our MIS algorithm—supported by Lemmas 5 and 7—provides exact error guarantees (probability=1), enabling optimal $k$-node selection as demonstrated in Section 6.
>
> We hope this addresses your concerns. Detailed explanations will be incorporated  in the revised manuscript.
>
> **W4: Readability & Intuition**
>
> We sincerely appreciate your valuable feedback. We regret that the extensive mathematical proofs may have compromised the paper's readability, despite their necessity for theoretical rigor. In the revised version, we will incorporate more intuitive explanations to enhance accessibility while maintaining mathematical precision. Thank you again for your constructive comments and for highlighting this important aspect of presentation.
>
> **Q1: Role of Sampling Algorithms**
>
> We sincerely appreciate your question. As you rightly pointed out, sampling algorithms identify approximate optimal sets for optimizing the overall equilibrium opinion, but their inherent stochastic nature precludes deterministic guarantees on the optimal node set. As previously noted, sampling methods typically offer more intuitive interpretations of the underlying mechanisms.
>
> In our work, the random-walk-based sampling algorithm served as conceptual inspiration for developing our asynchronous update framework. Regarding the forest sampling algorithm: since [27] originally proposed this method for a simplified model, and our model reduces to theirs under specific conditions (as demonstrated in our theoretical analysis), we extended their approach to our framework out of respect for prior work. This extension—coupled with its role as a baseline—enables direct comparison, further highlighting the efficiency and accuracy of our proposed asynchronous algorithm.
>
> We deeply value your attention to this issue and will meticulously revise the manuscript to strengthen the justification for sampling algorithms' inclusion, improve narrative coherence between methods, and enhance overall readability. Thank you again for your constructive feedback.
>
> **Limitations:**
>
> Our algorithm demonstrates clear advantages over sampling methods in both efficiency and accuracy, along with superior scalability. However, it is not without limitations. As formally established in Theorem 3, our approach remains sensitive to resistance coefficients—particularly exhibiting prolonged runtime when these coefficients are small. Furthermore, in exact optimization scenarios where $k_\text{gap}=0$ (i.e., boundary nodes contribute equally), the algorithm may fail to terminate gracefully, though such cases are practically negligible. To address this edge case, we propose mitigating the issue by relaxing constraint conditions. A dedicated discussion of limitations and future directions will be incorporated in the revised manuscript. Thank you again for your insightful suggestions.

---

### Official Review · Reviewer_vuNz · 2025-07-02

**Clarity:** 2
**Significance:** 2
**Originality:** 3
**Rating:** 4
**Confidence:** 4

**Summary:**

This paper studies the problem of maximizing opinions in social networks by modifying the internal opinions of key agents in a multi-agent network. The authors model opinion dynamics using a discrete-time linear system where each agent’s (time-varying) expressed opinion evolves based on its (fixed) internal opinion and the influence of others. At steady state, they derive a linear relationship between expressed and internal opinions, which forms the basis of their cost function measuring overall expressed opinion in the network. They analyze properties of this cost function and formulate the opinion maximization problem: selecting the top k agents whose internal opinions, if modified, would maximize overall expressed opinion. The authors propose sampling-based and asynchronous update-based methods to solve this problem and validate their approaches through comprehensive simulations on datasets including Google, Twitter, and DBLP.

**Questions:**

- There are many typing mistakes (e.g. “address” in line 2; “Equation” in line 112) and unclear sentences (e.g. “setting their expressed opinions…” in line 60); please correct them.
- Is the model below line 99 DeGroot or Friedkin–Johnsen? It seems like equation (1) comes from the steady-state behavior of z(t+1)=Az(t)+s(t), for some matrix A and constant vector s(t); adding intermediate equations before (1) would help.
- Please mention properties of f(R, s); is it non-negative, convex, and does Problem 1 fall under any known discrete or continuous optimization classes?
- The theoretical results and time complexity are presented in terms of network dimension (n), but the simulations only report actual implementation times. This disconnect is misleading; I suggest reporting results that reflect and validate the theoretical scaling with n.

**Ethical Concerns:**

["NO or VERY MINOR ethics concerns only"]

**Final Justification:**

I've not changed my score as the responses by the authors are not convincing to me.

**Limitations:**

The authors mention limitations only briefly, with just a throw-off sentence, which is inadequate. The paper would be stronger if they explained why they did not compare their methods to other existing techniques or consider inversion methods for sparse matrices. Additionally, in the checklist, the authors state that they discuss potential negative societal impact in Sections 1 and 8, but I could not find any such discussion; it would be helpful if they made this explicit.

**Quality:**

3

**Strengths And Weaknesses:**

Strengths: The paper is well written, clearly highlighting gaps in the literature and stating the problem with near mathematical precision. The algorithms and their time-complexity results, while not unexpected, are presented clearly and precisely. The theoretical results are supplemented with operational insights, emphasizing the probabilistic aspects and the role of network topology in the proposed algorithms. The experiments are thorough, covering networks from small to large scale to illustrate the time-complexity results effectively.

Weaknesses: I list four main weaknesses to justify my rating:

1) Despite the wealth of existing literature on opinion maximization, the authors unfortunately do not focus on how their algorithms compare with prior methods. This lack of fair comparison makes it hard to justify the merit of their research, in particular, how better are these methods (both qualitatively and qualitatively).
2) The authors’ basic premise is that standard matrix inversion methods to find top k influential nodes require O(n^3) time, which is prohibitive. While this claim is true in general, given the advances in sparse matrix inversion methods – and considering that the datasets used appear to be sparse (although I might be wrong) – it is surprising that the authors did not consider or compare against such approaches. This omission represents a second lack of fair comparison.
3) While the authors did a good job explaining the cost function in terms of central measures and other network parameters, the explanation of the optimization problem itself is weak. It does not clearly state what the optimization variables are or what it practically means to change internal opinions. Moreover, the role of network topology, such as the presence of cycles or other structural features, and how these affect optimization, is not adequately discussed. More focus is on the methods and tuning their performance than the problem itself.
4) In the experiments, except of the names of networks, there is limited information about the nature of the social networks and how they relate to the opinion dynamics motivation. There is no clear explanation of what expressed or internal opinions represent in these datasets. This raises the concern that similar results might hold on random networks or GPS networks or road networks, making it unconvincing whether the theoretical results have any practical (social) benefit. Including a real-world social network dynamics example would have made the validation more substantive.

---

> ### Author Rebuttal · Authors · 2025-07-30
>
> We sincerely appreciate you for your positive and constructive feedback.
>
> **W1: Existing Works**
>
> Extensive literature exists on the influence maximization problem, yet most works primarily focus on alternative optimization strategies—such as modifying node resistance coefficients or adding edges—while approaches based on adjusting initial opinions remain relatively scarce due to their inherent complexity. Although some existing studies explore initial opinion modification for opinion maximization, many are tailored to specific scenarios (e.g., leader-driven settings [26]).
>
> The work of [7] also addresses maximizing overall opinion through initial opinion adjustments but fundamentally relies on matrix inversion techniques. More recently, [27] proposed an approximate solution using forest sampling for this problem, employing matrix inversion as a baseline comparison. In our study, since the model incorporates the additional influence of resistance coefficients, we extend the method from [27] to accommodate our framework (Lines 185–202) and adopt it as a baseline for comparison. Notably, when fixing $\alpha_i = \frac{1}{1+d^+_i}$, our model naturally reduces to the formulations in [27] and [7]. Beyond forest sampling, we introduce an alternative approach based on absorbing random walks (Lines 157–184) and provide a comprehensive comparison between these methods.
>
> For all proposed algorithms, we rigorously analyze their time complexity. Although the bound for Algorithm 3 may appear loose due to necessary proof simplifications, it still guarantees near-linear time complexity. Experimental results demonstrate that our MIS algorithm not only runs substantially faster than competing methods but also achieves superior performance, highlighting its efficiency and effectiveness.
>
> **W2: Sparse Matrix Inversion**
>
> While [27] establishes matrix inversion as its baseline and demonstrates significant time efficiency improvements over this baseline, our extended adaptation of the method in [27] (as our baseline) similarly achieves superior performance compared to matrix inversion, which we have empirically verified.
>
> When fixing $\alpha_i = \frac{1}{1+d^+_i}$, our model naturally reduces to the foundational model in [27]. Given this theoretical equivalence and [27]'s already established performance advantages over matrix inversion, we prioritized comparative analysis against the approach in [27] rather than directly including matrix inversion as an additional baseline.
>
> Nevertheless, we acknowledge that practical sparse matrix inversion methods still exhibit suboptimal efficiency. As recommended, we will incorporate matrix inversion as an explicit baseline in the revised version to ensure comprehensive benchmarking. Thank you for your insightful feedback.
>
> **W3: Optimization Problem Clarity**
>
> Our optimization variables are the internal opinions $s_i$ of $k$ selected nodes (modified from their original values to 1), aimed at maximizing the overall equilibrium opinion. In practical applications, initial opinions typically originate from users' expressed opinion or predefined stance labels. Adjusting initial opinions effectively alters these nodes' intrinsic positions—such as converting neutral users into advocates—a common practice in marketing campaigns where persuading key users to shift product perceptions is critical.
>
> Regarding network topology, our rigorous theoretical proofs demonstrate that standard network characteristics (e.g., presence of cycles, directed/undirected edges, connectivity) do not compromise algorithmic correctness. This universality underscores our approach's robustness and scalability across diverse network structures, as empirically validated on real-world networks.
>
> We acknowledge that the manuscript overly emphasizes theoretical derivations at the expense of applied context. The revision will incorporate necessary practical explanations and case-based evidence to address this gap.
>
> **W4: Experimental Validation**
>
> The issues you raised are indeed critical, and we will address them through the following improvements:
>
> Regarding the insufficient dataset description, we will provide more comprehensive details about the datasets used. Specifically, the Twitter dataset represents a follower network capturing user follow relationships. In practical applications, initial opinions typically originate from users' expressed opinion or predefined stance labels, while the steady-state opinions reflect the final opinion distribution after social influence propagation. However, due to privacy constraints, publicly available datasets that simultaneously provide both network structures and ground-truth user opinions remain scarce, necessitating our use of generic networks. While our proposed algorithm conceptually applies to other network types (e.g., GPS or road networks), our research focus remains squarely on opinion dynamics within social networks. To strengthen validation, we plan to incorporate additional datasets with explicit opinion dynamics contexts—such as the real-world dataset from [33]—in the revised manuscript for more rigorous evaluation.
>
> Thank you again for your insightful suggestion.
>
> **Q1: Writing Quality**
>
> Thank you for your careful reading. We will check carefully the whole paper, correcting all grammatical errors and typos.
>
> **Q2: Opinion Dynamics Model**
>
> This model evolves from the DeGroot and Friedkin-Johnsen frameworks, initially introduced in [3]. With advancements in opinion dynamics research, additional variables—such as stubbornness coefficients and noise terms—have been incorporated to enhance realism. Strictly speaking, given its incorporation of self-reinforcement through initial opinions, this model falls under the Friedkin-Johnsen framework.
>
> We appreciate your insightful question. The revision will include:
>
> 1. Detailed explanations clarifying the model's origin and theoretical classification;
> 2. The explicit formulation $\boldsymbol{z}^{t+1} = \boldsymbol{R}\boldsymbol{s} + (\boldsymbol{I}-\boldsymbol{R})\boldsymbol{P}\boldsymbol{z}^t$ preceding Equation (1) to improve conceptual accessibility.
>
> **Q3: Properties of $f(\boldsymbol{R},\boldsymbol{s})$**
>
> As discussed in Lines 118-128, $f(\boldsymbol{R},\boldsymbol{s})$ represents a convex combination of all nodes' initial opinions with explicit non-negativity properties. When $\boldsymbol{R}$ is fixed, the function exhibits convexity, classifying Problem 1 as a discrete combinatorial optimization problem.
>
> **Q4: Time Complexity Validation**
>
> The time complexity analysis of MIS involves unavoidable bounding operations during the proof, resulting in a theoretical upper bound. In practical network scenarios, the empirical runtime consistently demonstrates significant improvements over this bound. To quantitatively demonstrate efficiency, we will supplement our analysis by tracking residual update counts, providing concrete metrics that directly reflect the algorithm's real-world efficiency and its alignment with the theoretical scaling with respect to $n$. Thank you again for your insightful suggestions.
>
> **Limitations:**
>
> Regarding limitations, our algorithm demonstrates clear advantages over sampling methods in both efficiency and accuracy, along with superior scalability. However, it is not without limitations. As formally established in Theorem 3, our approach remains influenced by resistance coefficients—particularly exhibiting prolonged runtime when these coefficients are small. Furthermore, in exact optimization scenarios where $k_\text{gap}=0$ (i.e., boundary nodes contribute equally), the algorithm may fail to terminate gracefully, though such cases are practically negligible. We will address this by proposing relaxed constraints as a mitigation strategy, with limitations and future directions discussed in the concluding section.
>
> As previously noted, our forest sampling algorithm extends the foundational work of [27]. To strengthen comparative analysis, we will incorporate matrix inversion benchmarks to highlight performance trade-offs.
>
> Concerning potential negative impacts, our method—designed to optimize overall opinions by modifying initial opinions—can yield positive societal benefits or commercial value. Nevertheless, it carries risks of opinion manipulation, including the propagation of harmful content [9]. We will prominently emphasize these dual-use considerations in the revised manuscript. Thank you for your suggestion!

---

> > ### Comment · Reviewer_vuNz · 2025-08-06
> >
> > Thank you for the detailed response. However, I found that the four weaknesses I raised were not addressed in a fully convincing manner. Most of the replies followed the general pattern of “we will address this in the revised manuscript,” which, while reasonable, raises the question of why preliminary investigations or proof-of-concept results weren’t included during the rebuttal phase. Such demonstrations could have reinforced the authors’ intent to follow through. Additionally, the responses to my (direct) points on weakness felt somewhat convoluted. I was hoping for clear and direct answers, something along the lines of “yes, because...” or “no, due to...”
> >
> > Finally, the authors noted that the "manuscript emphasizes theoretical derivations over applied context and intend to revise accordingly," I should clarify that some of my questions were also about theoretical aspects. If I understand correctly, the paper is meant to balance theoretical contributions with motivation from a practical problem. From that perspective, a fair evaluation requires attention to both theory and empirical validation, rather than emphasizing one at the expense of the other. Of course, this is my personal view, and I could be mistaken.

---

> ### Author Response · Authors · 2025-08-08
>
> Thank you for your feedback. Due to the large number of data nodes in our experiments, providing timely results for sparse matrix inversion proved challenging. As an alternative, we selected five small graph datasets from the same source as our working data to demonstrate our algorithm's advantages. Using LU decomposition with forward/backward substitution as our baseline for matrix inversion, our MIS algorithm showed significantly faster runtime performance (see results below).
> | name         | nodes  | edges   | type | LU+forward/backward substitution(s) | MIS(s) |
> | ------------ | ------ | ------- | ---- | --------------------------------------- | ------ |
> | ego-facebook | 4,039   | 88,234  | U    | 1.59                                    | 0.20   |
> | wiki-vote    | 7,115   | 103,689 | U    | 12.86                                   | 0.21   |
> | dblp-cite    | 12,590 | 49,759  | D    | 1.98                                    | 0.27   |
> | ego-twitter  | 23,370 | 33,101  | D    | 6.01                                    | 0.22   |
> | Brightkite   | 58,228 | 214,078 | U    | 558.11                                  | 0.24   |
>
> We further analyzed our algorithm's time complexity by tracking residual vector updates relative to node count $n$. The results demonstrate our algorithm achieves near ​​$O(n)​$​ time complexity in practice, with a constant factor ($\le 300$) that remains stable regardless of network size. Additionally, our algorithm maintains an $O(n)$ space complexity, making it highly scalable for large-scale networks. This represents a substantial improvement over even state-of-the-art sparse matrix inversion methods.
>
> | networks    | updates | updates/n |
> | ----------- | ------- | --------- |
> | DBLP        | 1.58e7  | 49.93     |
> | Google      | 2.37e7  | 27.06     |
> | YoutubeSnap | 5.60e7  | 49.34     |
> | Pokec       | 2.07e8  | 126.86    |
> | Flixster    | 2.89e8  | 114.84    |
> | LiveJournal | 5.14e8  | 106.05    |
> | Twitter     | 1.03e10 | 249.28    |
> | SinaWeibo   | 4.62e9  | 78.77     |
>
> While we attempted to test on real-world FJ-model networks, we found the dataset from [33] wasn't fully available. To comply with conference policies, we'll contact the authors after rebuttal. Based on our comprehensive experiments, we're confident our algorithm would perform well on such networks.
>
> We sincerely apologize if our response caused any confusion—please don't hesitate to raise any remaining questions so we may provide better clarification. Thank you for your time and help.

---

### Official Review · Reviewer_dkDZ · 2025-07-05

**Clarity:** 2
**Significance:** 2
**Originality:** 3
**Rating:** 4
**Confidence:** 2

**Summary:**

This paper proposes opinion maximization algorithms for linear dynamical system models for opinion dynamics. The authors build heavily on a previous paper reference [27] which is also for the same purpose. The authors propose an approximate algorithm for influence estimation algorithm 2 and then the max influence selector which is algorithm 3. The authors show results on Benchmark datasets.

**Questions:**

What is the speedup in terms of time complexity?

How exactly is this achieved, over say Forest? Showing this with some small networks will help.

What is the performance hit?

**Ethical Concerns:**

["NO or VERY MINOR ethics concerns only"]

**Final Justification:**

I still don't buy the argument that Degroot model is older and hence SLANT can't be compared against. According to my understanding the varying susecptibility model (reference [3]) is also a linear model, and hence can be controlled through linear dynamical systems. It is important that a comparison be made with this class of methods.

About the experimental results in the performance, the graphs of MIS and Forest are overlapping. Hence, my confusion. Something should be done to improve the readability of that graph.

**Limitations:**

There is no limitations section

**Quality:**

3

**Strengths And Weaknesses:**

Strengths:
- The paper is clearly written, with some claimed algorithmic contributions
- The problem itself is not very  novel but relevant.

Weaknesses:
- The main weakness is that the advantage of the proposed method over the baseline e.g. [27] is not clear. Algorithm 1 is from the baseline. How is algorithm 2 different? What is the time complexity of the proposed algorithm?

- The second big weakness is that the paper does not connect very well to the existing literature. I think the basic problem of controlling linear dynamical systems has been tackled in great detail in the literature. Specifically in the context of opinion dynamics, see:
https://dl.acm.org/doi/10.5555/3237383.3237899
How is the present algorithm better than the state of the art in that literature?

- Finally, in experimental results it looks like MIS performs much more poorly than Forest.

---

> ### Author Rebuttal · Authors · 2025-07-30
>
> Thank you for the feedback and suggestions. Below are our response to your main concerns.
>
> **W1: Algorithm Advantages**
>
> The baseline sampling algorithm relies on probabilistic approximation, meaning it can only provide probabilistic error guarantees and thus obtains an approximate optimal node set. According to Chernoff bound, its required sample size is directly proportional to $\epsilon^{-2}$, which highlights its inherent limitations. In contrast, our algorithm employs deterministic derivation to provide exact error guarantees, enabling the identification of the precise optimal node set. As formally stated in Theorem 3, the time complexity of our algorithm MIS scales with  $\epsilon$ as $\log{(1/\epsilon)}$, making it robust to the choice of error parameter $\epsilon$ and ensuring superior efficiency. Experimental results in Table 1 show that our MIS algorithm consistently outperforms baseline methods in computational efficiency across all scenarios, while Figure 1 demonstrates its better performance quality, with additional advantages detailed in Appendix C.
>
> Our Algorithm 1 is inspired by absorbing random walk-based sampling methods, using asynchronous deterministic updates to screen candidate nodes from the entire graph. Algorithm 2 then performs reverse-direction asynchronous updates from individual node perspectives to refine this candidate set, ultimately converging to the exact optimal solution. The correctness of our approach is guaranteed by Lemma 8. The time complexity of MIS is formally presented in Theorem 3, with detailed complexity analysis for both Algorithm 1 and Algorithm 2 provided in its proof. Notably, the proof involves necessary bounding operations, meaning Theorem 3 provides an upper bound on the actual time complexity—our implementation consistently achieves better performance than this theoretical guarantee.
>
> In summary, by leveraging deterministic asynchronous propagation methods and the mechanism formalized in Lemma 8, our algorithm efficiently obtains the exact optimal solution for Problem 1. MIS demonstrates significant advantages over baseline methods (RWB and Forest) in both computational efficiency and solution quality. Thank you.
>
> **W2: Literature Comparison**
>
> We appreciate the reviewer’s feedback regarding the connection between our work and existing literature. While we acknowledge the foundational contributions of their work in the field of linear dynamical system control, we maintain that our contribution remains significant due to fundamental differences in models, research objectives, and scalability. Below, we provide a detailed clarification:
>
> 1. **Model Differences**: The two papers employ distinct models—one uses the SLANT model, while ours is the model introduced in [3]. To our knowledge, the authors of SLANT only compared their model with the original DeGroot model, which has since undergone extensive developments (e.g., incorporating stubbornness coefficients, noise, etc.). Thus, it remains unclear whether SLANT is superior to the model we adopt.
> 2. **Research Focus**: Our work aims to develop fast and efficient algorithms for solving the target problem, with our approach scalable to networks of tens of millions of nodes even in a single-threaded setting. In contrast, the referenced work primarily discusses a feasibility framework under the SLANT model, with limited attention to algorithmic design, computational complexity, or practical runtime. The largest real-world network they evaluated contains only 1,031 nodes, making it difficult to assess the scalability and real-world applicability of their method.
> 3. **Target Set Differences**: As shown in their Figure 3, the target set in their work overlaps significantly with sets derived from common centrality measures. In our case, however, experimental results demonstrate that our target set differs substantially from those obtained via common centrality metrics. This indicates that our work pursues a distinct objective, potentially leading to different practical implications.
> 4. **Theoretical and Practical Foundations**: As highlighted in our related work section, FJ-related models have recently seen widespread research and applications, further underscoring the value of our contributions.
>
> Nevertheless, we will highlight the contributions of their work in the related work section to further ensure the comprehensiveness of our paper.
>
> **W3: Experimental Performance**
>
> We appreciate the reviewer's attention to our experimental results. Regarding the comparison between MIS and Forest algorithms, we would like to clarify that in all tested scenarios, our algorithm MIS demonstrates consistently superior performance over Forest.
>
> Specifically, as shown in Table 1, MIS achieves better computational efficiency than Forest across all test cases, with order-of-magnitude improvements in most scenarios. The results in Figure 1 further confirm that under identical experimental configurations, MIS not only maintains optimal performance but also achieves perfect accuracy (with precision metrics consistently at 1). Particularly in the large-scale graph tests shown in Figure 7 (with resistance coefficients following exponential distributions), while Forest exhibits noticeable difficulties in identifying optimal solutions, our MIS algorithm reliably produces perfectly accurate optimal node sets.
>
>  In summary, our algorithm MIS significantly outperforms the Forest algorithm in terms of efficiency and accuracy. Thank you.
>
> **Q1: Time Complexity**
>
> As shown in Theorem 3, our algorithm has a near-linear time complexity bound. Due to necessary proof simplifications, real-world performance is significantly better than this bound. Based on the suggestion of Reviewer vuNz, we will measure actual performance via residual update statistics to further demonstrate its efficiency advantages.
>
> **Q2: Simple Example**
>
> Our algorithm achieves its correctness guarantee through Lemma 8, which relies on the error bounds established in Lemmas 5 and 7. Based on this analysis, the MIS algorithm can compute exact optimal solutions with significantly higher efficiency compared to Forest and other sampling-based methods.
>
> Consider a two-node graph with adjacency matrix $\boldsymbol{A} = \begin{pmatrix}0&1\\\1&0\end{pmatrix}$, stubbornness matrix $\boldsymbol{R} = \begin{pmatrix}0.5&0\\\0&0.5\end{pmatrix}$, and initial opinion vector $\boldsymbol{s} = (0.8, 0.6)^\top$. With $k=1$ and $\epsilon=0.5$ for Algorithm 1, the updates to $(\tilde{\boldsymbol{\Delta}}, \boldsymbol{r}_a)$ proceed as follows:
>
> | updates | node 1     | node 2      | queue       |
> | ------- | ---------- | ----------- | ----------- |
> | 0       | (0,1)      | (0,1)       | {1,2}       |
> | 1       | (0.1,0)    | (0,1.5)     | {2}         |
> | 2       | (0.1,0.75) | (0.3,0)     | {1}         |
> | 3       | (0.175,0)  | (0.3,0.375) | $\emptyset$ |
>
> The algorithm terminates after 3 updates when $\epsilon=0.5$, producing $\tilde{\boldsymbol{\Delta}}=(0.175,0.3)^\top$ and $\boldsymbol{r}_a = (0, 0.375)^\top$. Applying Lemmas 5 and 8 gives $T = \\{v| \tilde{\boldsymbol{\Delta}}(v) \ge 0.35\\} = \emptyset$ and $C = \\{v | 0.35 > \tilde{\boldsymbol{\Delta}}(v) \ge 0.15\\} = \\{1,2\\}$. In practice, $C$ typically contains very few nodes - for Twitter's 10M+ node network we observe $|C|<100$.
>
> For convenience, we execute lines 5-11 of Algorithm 3 with $\epsilon' = 0.375/2$  (where $\epsilon'/(\boldsymbol{r}_a) _{\text{sum}} = 0.5$). Since no nodes in $C$ satisfy the starting condition of Algorithm 2, the vector $\tilde{\boldsymbol{\Delta}}$ remains unchanged. By Lemmas 7 and 8, $T = \\{v| \tilde{\boldsymbol{\Delta}}(v) \ge 0.175 + (0.375)^2/2\\} = \\{2\\}$, and the algorithm terminates with the exact solution. This example demonstrates MIS requires just 3 updates for precise optimization, while Forest would need far more samples to achieve close precision. Both theoretical analysis and experimental results confirm the substantial advantage of MIS over Forest.
>
> **Q3: Algorithm Performance**
>
> As stated above, we have demonstrated that Algorithm MIS achieves a near-linear time complexity upper bound. Experimental results confirm that MIS significantly outperforms baseline algorithms in both efficiency and accuracy. Thank you for your time and help.

---

### Decision · Program_Chairs · 2025-09-17

**Decision:**

Accept (poster)

**Comment:**

The paper focus on maximizing opinions of users by leveraging Degroot model and linear dynamical system and designing a control system on top of it. They provide asynchronous update, which mimics reality and evaluate their method against several baselines, which shows that their method performs better.

The reviewers are generally positive about the paper. However, reviewer dkDz pointed out some related work and commented on novelty. I think the problem is not novel, however, their approach is novel and clearly placed in the paper. I recommend acceptance.